# Primal Dual Interpretation of the Proximal Stochastic Gradient Langevin Algorithm

**Adil Salim**     **Peter Richtárik**
King Abdullah University of Science and Technology, Thuwal, Saudi Arabia

## Abstract

We consider the task of sampling with respect to a log concave probability distribution. The potential of the target distribution is assumed to be composite, *i.e.*, written as the sum of a smooth convex term, and a nonsmooth convex term possibly taking infinite values. The target distribution can be seen as a minimizer of the Kullback-Leibler divergence defined on the Wasserstein space (*i.e.*, the space of probability measures). In the first part of this paper, we establish a strong duality result for this minimization problem. In the second part of this paper, we use the duality gap arising from the first part to study the complexity of the Proximal Stochastic Gradient Langevin Algorithm (PSGLA), which can be seen as a generalization of the Projected Langevin Algorithm. Our approach relies on viewing PSGLA as a primal dual algorithm and covers many cases where the target distribution is not fully supported. In particular, we show that if the potential is strongly convex, the complexity of PSGLA is $\mathcal{O}(1/\varepsilon^2)$ in terms of the 2-Wasserstein distance. In contrast, the complexity of the Projected Langevin Algorithm is $\mathcal{O}(1/\varepsilon^{12})$ in terms of total variation when the potential is convex.

## 1   Introduction

Sampling from a target distribution is a fundamental task in machine learning. Consider the Euclidean space $\mathsf{X} = \mathbb{R}^d$ and a convex function $V : \mathsf{X} \to (-\infty, +\infty]$. Assuming that $\exp(-V)$ has a positive finite integral w.r.t. the Lebesgue measure $\mathrm{Leb}$, we consider the task of sampling from the distribution $\mu^\star$ whose density is proportional to $\exp(-V(x))$ (we shall write $\mu^\star \propto \exp(-V)$).

If $V$ is smooth, Langevin algorithm produces a sequence of iterates $(x^k)$ asymptotically distributed according to a distribution close to $\mu^\star$. Langevin algorithm performs iterations of the form

$$x^{k+1} = x^k - \gamma \nabla V(x^k) + \sqrt{2\gamma} W^{k+1}, \tag{1}$$

where $\gamma > 0$ and $(W^k)_k$ is a sequence of i.i.d. standard Gaussian vectors in $\mathsf{X}$. Each iteration of (1) can be seen as a gradient descent step for $V$, where the gradient of $V$ is perturbed by a Gaussian vector. Hence, the iterations of Langevin algorithm look like those of the stochastic gradient algorithm; however the noise in Langevin algorithm is scaled by $\sqrt{\gamma}$ instead of $\gamma$. Nonasymptotic bounds for Langevin algorithm have been established in [17, 20]. Moreover, Langevin algorithm can be interpreted as an inexact gradient descent method to minimize the Kullback-Leibler (KL) divergence w.r.t. $\mu^\star$ in the space of probability measures [1, 5, 14, 19, 23, 33].

In many applications, the function $V$ is naturally written as the sum of a smooth and a nonsmooth term. In Bayesian statistics for example, $\mu^\star$ typically represents some posterior distribution. In this case, $V$ is the sum of the $\log$-likelihood (which is itself a sum over the data points) and the possibly nonsmooth potential of the prior distribution [19, 21, 32], which plays the role of a regularizer. In some other applications in Bayesian learning, the support of $\mu^\star$ is not the whole space $\mathsf{X}$ [9, 10] (*i.e.*, $V$ can take the value $+\infty$). In order to cover these applications, we consider the case where $V$ is

written as

$$V(x) := \mathbb{E}_{\xi}(f(x, \xi)) + G(x), \tag{2}$$

where $\xi$ is a random variable, $f(\cdot, s) : \mathsf{X} \to \mathbb{R}$ for every $s \in \Xi$, $F(x) = \mathbb{E}_{\xi}(f(x, \xi))$ is smooth and convex and $G : \mathsf{X} \to (-\infty, +\infty]$ is nonsmooth and convex. We assume to have access to the stochastic gradient $\nabla_x f(x, \xi)$ (where $\xi$ is a random variable with values in $\Xi$) and to the proximity operator $\mathrm{prox}_{\gamma G}$ of $G$. The template (2) covers many log concave densities [10, 13, 19, 21]. In optimization, the minimization of $V$ can be efficiently tackled by the proximal stochastic gradient algorithm [3]. Inspired by this optimization algorithm, the Proximal Stochastic Gradient Langevin Algorithm (PSGLA) [19] is the method performing proximal stochastic gradient Langevin steps of the form

$$x^{k+1} = \mathrm{prox}_{\gamma G}\left(x^k - \gamma \nabla_x f(x^k, \xi^{k+1}) + \sqrt{2\gamma} W^{k+1}\right), \tag{3}$$

where $\gamma > 0$, $(W^k)$ is a sequence of i.i.d. standard Gaussian random vectors in $\mathsf{X}$, and $(\xi^k)$ is a sequence of i.i.d. copies of $\xi$. Remarkably, the iterates $x^k$ of PSGLA remain in the domain of $G$, *i.e.*, the support of $\mu^\star$, a property that is useful in many contexts. When $G$ is Lipschitz continuous, the support of $\mu^\star$ is $\mathsf{X}$ and PSGLA can be interpreted as an inexact proximal gradient descent method for minimizing KL, with convergence rates proven in terms of the KL divergence [19]. However, for general $G$, the KL divergence can take infinite values along PSGLA. Therefore, a new approach is needed.

## 1.1 Related works

**First, various instances of the PSGLA algorithm have already been considered.** The only instance allowing $G(x)$ to be infinite (*i.e.*, the support of $\mu^\star$ not to be $\mathsf{X}$) is the Projected Langevin Algorithm [10], which corresponds to our setting in the special case with $G = \iota_C$ (*i.e.*, the indicator function of a convex body[1] $C$), and $\nabla f(\cdot, s) \equiv \nabla F$ for every $s$ (*i.e.*, the full gradient of $F$). In this case, $\mathrm{prox}_{\gamma G}$ is the orthogonal projection onto $C$ and $\mu^\star$ is supported by $C$. Bubeck et al [10] provide complexity results in terms of sufficient number of iterations to achieve $\varepsilon$ accuracy in terms of the Total Variation between the target distribution $\mu^\star$ and the current iterate distribution. Assuming that $F$ is convex and smooth, the complexity of the Projected Langevin Algorithm is $\mathcal{O}(1/\varepsilon^{12})^2$, and if $F \equiv 0$, the complexity is improved to $\mathcal{O}(1/\varepsilon^8)$.

Other instances of PSGLA were proposed in the case where $G$ is Lipschitz continuous or smooth (and hence finite). Wibisono [33] considered the case with $F = G$ and $\nabla f(\cdot, s) \equiv \nabla F$, proposing the Symmetrized Langevin Algorithm (SLA), and showed that the current iterate distribution converges linearly in Wasserstein distance to the invariant measure of the SLA, if $F$ is strongly convex and smooth. Durmus et al [19] considered the case where $G$ is Lipschitz continuous, and showed that the complexity of PSGLA is $\mathcal{O}(1/\varepsilon^2)$ in terms of the KL divergence and $\mathcal{O}(1/\varepsilon^4)$ in terms of the Total Variation distance if $F$ is convex and smooth. If $F$ is strongly convex, the complexity is $\mathcal{O}(1/\varepsilon^2)$ in Wasserstein distance and $\mathcal{O}(1/\varepsilon)$ in KL divergence. Bernton [5] studied a setting similar to [19] and derived a similar result for the Proximal Langevin Algorithm (*i.e.*, PSGLA without the gradient step) in the strongly convex case. The Proximal Langevin Algorithm was also studied in a recent paper of Wibisono [34], where a rapid convergence result was proven in the case where $G$ is nonconvex but satisfies further smoothness and geometric assumptions.

**Second, the task of sampling w.r.t. $\mu^\star$, where $G$ is nonsmooth and possibly takes infinite values, using Langevin algorithm, has also been considered.** When $F$ is strongly convex and $G$ an indicator function of a bounded convex set, the existence of an algorithm achieving $\mathcal{O}(1/\varepsilon^2)$ in Wasserstein and Total Variation distances was proven by Hsieh et al [22, Theorem 3]. However, an actual algorithm is only given in a specific, although nonconvex, case. Besides, MYULA (Moreau-Yosida Unadjusted Langevin Algorithm) [9, 21] can tackle the task of sampling from $\mu^\star$ efficiently. MYULA is equivalent to Langevin algorithm (1) applied to sampling from $\mu^\lambda \propto \exp(-F - G^\lambda)$, where $G^\lambda$ is the Moreau-Yosida approximation of $G$ [2]. By choosing the smoothing parameter $\lambda > 0$ appropriately, and making assumptions that allow to control the distance between $\mu^\lambda$ and $\mu^\star$ (*e.g.*, $G$ Lipschitz or $G = \iota_C$), complexity results for MYULA were established in [9, 21]. For example, if $G$ is the indicator function of a convex body, Brosse et al [9] show that the complexity

of MYULA is $\mathcal{O}(1/\varepsilon^6)$ in terms of the Total Variation distance (resp. 1-Wasserstein distance) if $F$ is convex and smooth (resp., if $F$ is strongly convex and smooth), provided that the algorithm is initialized from a minimizer of $V$. Similarly to PSGLA, MYULA involves one proximal step and one gradient step per iteration. However, the support of the smoothed distribution $\mu^\lambda$ is always $\mathsf{X}$ (even if $\mu^\star$ is not fully supported), and therefore the iterates of MYULA do not remain in the support of the target distribution $\mu^\star$, contrary to PSGLA.

**Finally, the task of sampling w.r.t. $\mu^\star$, where $V$ is not smooth but finite, has also been considered.** The Perturbed Langevin Algorithm proposed by Chatterji et al [12] allows to sample from $\mu^\star$ in the case when $G$ satisfies a weak form of smoothness (generalizing both Lipschitz continuity and smoothness) and without accessing its proximity operator. Finally, if $G$ is Lipschitz continuous, the Stochastic Proximal Langevin Algorithm proposed by Salim et al [27] and Schechtman et al [28] allows to sample from $\mu^\star$ using cheap stochastic proximity operators only.

## 1.2 Contributions

In summary, PSGLA has complexity $\mathcal{O}(1/\varepsilon^2)$ in 2-Wasserstein distance if $F$ is strongly convex [19] and $G$ is Lipschitz. The only instance of PSGLA allowing $G$ to be infinite is the Projected Langevin Algorithm. It has complexity $\mathcal{O}(1/\varepsilon^{12})$ in Total Variation [10][3] and only applies to the case where $G$ is the indicator of a convex body. In the latter case, another Langevin algorithm called MYULA has complexity $\mathcal{O}(1/\varepsilon^6)$ in 1-Wasserstein distance [9], but allows the iterates to leave the support of $\mu^\star$. Besides, still in the case where $G$ is an indicator function, there exists a Langevin algorithm achieving $\mathcal{O}(1/\varepsilon^2)$ rate in the Wasserstein distance [22].

*In this paper, we consider other (i.e., new) cases where $G$ can take infinite values. More precisely, we consider a general nonsmooth convex function $G$ and we assume that $\exp(-V)$ has a mild Sobolev regularity. We develop new mathematical tools (e.g., a Lagrangian for the minimization of KL), that have their own interest, to obtain our complexity results. Our main result is to show that, surprisingly, PSGLA still has the complexity $\mathcal{O}(1/\varepsilon^2)$ in 2-Wasserstein distance if $F$ is strongly convex, although $G$ can take infinite values. We also show that, if $F$ is just convex, PSGLA has the complexity $\mathcal{O}(1/\varepsilon^2)$ in terms of a newly defined duality gap, which can be seen as the notion that replaces KL, since KL can be infinite.*

Our approach follows the line of works [5, 14, 19, 23, 25, 31, 33, 34] that formulate the task of sampling form $\mu^\star$ as the problem of minimizing the KL divergence w.r.t $\mu^\star$. In summary, our contributions are the following.

• In the first part of the paper, we reformulate the task of sampling from $\mu^\star$ as the resolution of a monotone inclusion defined on the space of probability measures. We subsequently use this reformulation to define a duality gap for the minimization of the KL divergence, and show that strong duality holds.

• In the second part of this paper, we use this reformulation to represent PSGLA as a primal dual stochastic Forward Backward algorithm involving monotone operators.

• This new representation of PSGLA, along with the strong duality result from the first part, allows us to prove new complexity results for PSGLA that extend and improve the state of the art.

• Finally, we conduct some numerical experiments for sampling from a distribution supported by a set of matrices (see appendix).

In the first part we combine tools from optimization duality [16] and optimal transport [1] and in the second part we combine tools from the analysis of the Langevin algorithm [19], and the analysis of primal dual optimization algorithms [11, 18].

The remainder is organized as follows. In Section 2 we provide some background knowledge on convex analysis and optimal transport. In Section 3 we develop a primal dual optimality theory for the task of sampling from $\mu^\star$. In Section 4 we give a new representation of PSGLA using monotone operators. We use it to state our main complexity result on PSGLA in Section 5. Numerical experiments and all proofs are postponed to the appendix. Therein, we also provide further intuitions on PSGLA, namely the connection between gradient descent and Langevin algorithm [19] and the

connection between primal dual optimization and our approach. Finally, an extension of PSGLA for handling a third (stochastic, Lipschitz continuous and proximable) term in the definition of the potential $V$ (2) is provided at the end of the appendix.

## 2  Background

Throughout this paper, we use the conventions $\exp(-\infty) = 0$ and $1/0 = +\infty$.

### 2.1  Convex analysis

In this section, we recall some facts from convex analysis. These facts will be used in the proofs without mention. For more details, the reader is referred to [4].

#### 2.1.1  Convex optimization

By $\Gamma_0(\mathsf{X})$ we denote the set of proper, convex, lower semicontinuous functions $\mathsf{X} \to (-\infty, +\infty]$. A function $F \in \Gamma_0(\mathsf{X})$ is $L$-smooth if $F$ is differentiable and its gradient $\nabla F$ is $L$-Lipschitz continuous. Consider $G \in \Gamma_0(\mathsf{X})$ and denote $\mathrm{dom}(G) := \{x \in \mathsf{X} \ : \ G(x) < \infty\}$ its domain. Given $x \in \mathsf{X}$, a subgradient of $G$ at $x$ is any vector $y \in \mathsf{X}$ satisfying

$$G(x) + \langle y, x' - x \rangle \leq G(x'), \tag{4}$$

for every $x' \in \mathsf{X}$. If the set $\partial G(x)$ of subgradients of $G$ at $x$ is not empty, then there exists a unique element of $\partial G(x)$ with minimal norm. This particular subgradient is denoted $\partial^0 G(x)$. The set valued map $\partial G(\cdot)$ is called the subdifferential. The proximity operator of $G$, denoted $\mathrm{prox}_G$, is defined by

$$\mathrm{prox}_G(x) := \arg\min_{x' \in \mathsf{X}} \left\{ G(x') + \tfrac{1}{2}\|x - x'\|^2 \right\}. \tag{5}$$

By $\iota_C(\cdot)$ we denote the indicator function of set $C$ given by $\iota_C(x) = 0$ if $x \in C$ and $\iota_C(x) = +\infty$ if $x \notin C$. If $G = \iota_C$, where $C$ is a closed convex set, then $\mathrm{prox}_G$ is the orthogonal projection onto $C$. Moreover, $\mathrm{prox}_G(x)$ is the only solution $x'$ to the inclusion $x \in x' + \partial G(x')$. The Fenchel transform of $G$ is the function $G^* \in \Gamma_0(\mathsf{X})$ defined by $G^*(y) := \sup_{x \in \mathsf{X}} \{\langle y, x \rangle - G(x)\}$. Several properties relate $G$ to its Fenchel transform $G^*$. First, the Fenchel transform of $G^*$ is $G$. Then, the subdifferential $\partial G^*$ is characterized by the relation $x \in \partial G^*(y) \Leftrightarrow y \in \partial G(x)$. Finally, $G^*$ is $\lambda$-strongly convex if and only if $G$ is $1/\lambda$-smooth.

#### 2.1.2  Maximal monotone operators

A set valued function $A : \mathsf{X} \rightrightarrows \mathsf{X}$ is *monotone* if $\langle y - y', x - x' \rangle \geq 0$ whenever $y \in A(x)$ and $y' \in A(x')$. The *inverse* of $A$, denoted $A^{-1}$, is defined by the relation $x \in A^{-1}(y) \Leftrightarrow y \in A(x)$, and the *set of zeros* of $A$ is $Z(A) := A^{-1}(0)$. If $A$ is monotone, $A$ is *maximal* if its *resolvent*, *i.e.*, the map $J_A : x \mapsto (I + A)^{-1}(x)$, is single valued. If $G \in \Gamma_0(\mathsf{X})$, then $\partial G$ is a maximal monotone operator and $J_{\partial G} = \mathrm{prox}_G$. Moreover, $Z(\partial G) = \arg\min G$ and $(\partial G)^{-1} = \partial G^*$. If $S$ is a skew symmetric matrix on $\mathsf{X}$, the operator $x \mapsto Sx$ is maximal monotone. Finally, the sum $\partial G + S$ is also a maximal monotone operator. Many problems in optimization can be cast as the problem of finding a zero $x$ of the sum of two maximal monotone operators $0 \in (A + B)(x)$ [16]. For instance, $Z(\nabla F + \partial G) = \arg\min F + G$. To solve this problem, the Forward Backward algorithm is given by the iteration $x^{k+1} = J_{P^{-1}A}(x^k - P^{-1}B(x^k))$, where $P$ is a symmetric positive definite matrix $(P \in \mathbb{R}^{d \times d}_{++})$,[4] and $B$ is single valued. Using the definition of the resolvent, the Forward Backward algorithm can equivalently be written as

$$P(x^{k+1/2} - x^k) = -\gamma B(x^k), \quad P(x^{k+1} - x^{k+1/2}) \in -\gamma A(x^{k+1}). \tag{6}$$

### 2.2  Optimal transport

In this section, we recall some facts from optimal transport theory. These facts will be used in the proofs without mention. For more details, the reader is referred to Ambrosio et al [1].

### 2.2.1 Wasserstein distance

By $\mathscr{B}(\mathsf{X})$ we denote the $\sigma$-field of Lesbesgue measurable subsets of $\mathsf{X}$, and by $\mathcal{P}_2(\mathsf{X})$ the set of probability measures $\mu$ over $(\mathsf{X}, \mathscr{B}(\mathsf{X}))$ with finite second moment $\int \|x\|^2 d\mu(x) < \infty$. Denote $\mathrm{supp}(\mu)$ the support of $\mu$. The identity map $I$ belongs to the Hilbert space $L^2(\mu; \mathsf{X})$ of $\mu$-square integrable random vectors in $\mathsf{X}$. We denote $\langle \cdot, \cdot \rangle_\mu$ (resp. $\| \cdot \|_\mu$) the inner product (resp. the norm) in this space. Given $T : \mathsf{X} \to \mathsf{Z}$, where $\mathsf{Z}$ is some Euclidean space, the *pushforward measure* of $\mu$ by $T$, also called the *image measure*, is defined by $T \# \mu(A) := \mu(T^{-1}(A))$ for every $A \in \mathscr{B}(\mathsf{Z})$. Consider $\mu, \nu \in \mathcal{P}_2(\mathsf{X})$. A *coupling* $\upsilon$ between $\mu$ and $\nu$ (we shall write $\upsilon \in \Gamma(\mu, \nu)$) is a probability measure over $(\mathsf{X}^2, \mathscr{B}(\mathsf{X}^2))$ such that $x^\star \# \upsilon = \mu$, where $x^\star : (x, y) \mapsto x$, and $y^\star \# \upsilon = \nu$, where $y^\star : (x, y) \mapsto y$. In other words, $(X, Y)$ is a random variable such that the distribution of $X$ is $\mu$ (we shall write $X \sim \mu$) and $Y \sim \nu$ if and only if the distribution of $(X, Y)$ is a coupling. The (2-)*Wasserstein distance* is then defined by

$$W^2(\mu, \nu) := \inf_{\upsilon \in \Gamma(\mu, \nu)} \int \|x - y\|^2 d\upsilon(x, y). \tag{7}$$

Let $\mathcal{P}_2^r(\mathsf{X})$ be the set of elements $\mu \in \mathcal{P}_2(\mathsf{X})$ such that $\mu$ is absolutely continuous w.r.t. Leb (we shall write $\mu \ll \mathrm{Leb}$). Brenier's theorem asserts that if $\mu \in \mathcal{P}_2^r(\mathsf{X})$, then the $\inf$ defining $W^2(\mu, \nu)$ is actually a $\min$ achieved by a unique minimizer $\upsilon$. Moreover, there exists a uniquely determined $\mu$-almost everywhere (a.e.) map $T_\mu^\nu : \mathsf{X} \to \mathsf{X}$ such that $\upsilon = (I, T_\mu^\nu) \# \mu$, where $(I, T_\mu^\nu) : x \mapsto (x, T_\mu^\nu(x))$. In this case, $T_\mu^\nu$ is called the *optimal pushforward* from $\mu$ to $\nu$ and satisfies

$$W^2(\mu, \nu) = \int \|x - T_\mu^\nu(x)\|^2 d\mu(x). \tag{8}$$

### 2.2.2 Geodesically convex functionals

We shall consider several functionals defined on the space $\mathcal{P}_2(\mathsf{X})$. For every $\mu \in \mathcal{P}_2^r(\mathsf{X})$ with density denoted $\mu(x)$ w.r.t. Leb, the *entropy* is defined by

$$\mathcal{H}(\mu) := \int \log(\mu(x)) d\mu(x), \tag{9}$$

and if $\mu \notin \mathcal{P}_2^r(\mathsf{X})$, then $\mathcal{H}(\mu) := +\infty$. Given $V \in \Gamma_0(\mathsf{X})$, the *potential energy* is defined for every $\mu \in \mathcal{P}_2(\mathsf{X})$ by

$$\mathcal{E}_V(\mu) := \int V(x) d\mu(x). \tag{10}$$

Finally, if $\mu' \in \mathcal{P}_2(\mathsf{X})$ such that $\mu \ll \mu'$, the *Kullback-Leibler (KL) divergence* is defined by

$$\mathrm{KL}(\mu | \mu') := \int \log \left( \tfrac{d\mu}{d\mu'}(x) \right) d\mu(x), \tag{11}$$

where $\frac{d\mu}{d\mu'}$ denotes the density of $\mu$ w.r.t. $\mu'$, and $\mathrm{KL}(\mu | \mu') := +\infty$ if $\mu$ is not absolutely continuous w.r.t. $\mu'$. The functionals $\mathcal{H}, \mathcal{E}_V$ and $\mathrm{KL}(\cdot | \mu^\star)$ satisfy a form of convexity over $\mathcal{P}_2(\mathsf{X})$ called *geodesic convexity*. If $\mathcal{F} : \mathcal{P}_2(\mathsf{X}) \to (-\infty, +\infty]$ is geodesically convex, then for every $\mu \in \mathcal{P}_2^r(\mathsf{X})$, $\mu' \in \mathcal{P}_2(\mathsf{X})$, and $\alpha \in [0, 1]$, $\mathcal{F}\left( (\alpha T_\mu^{\mu'} + (1 - \alpha)I) \# \mu \right) \le \alpha \mathcal{F}(\mu') + (1 - \alpha)\mathcal{F}(\mu)$. Given $\mu \in \mathcal{P}_2^r(\mathsf{X})$, a *(Wasserstein) subgradient* of $\mathcal{F}$ at $\mu$ is a random variable $Y \in L^2(\mu; \mathsf{X})$ such that for every $\mu' \in \mathcal{P}_2(\mathsf{X})$,

$$\mathcal{F}(\mu) + \langle Y, T_\mu^{\mu'} - I \rangle_\mu \le \mathcal{F}(\mu'). \tag{12}$$

Moreover, if $Y'$ is a subgradient of $\mathcal{F}$ at $\mu'$, then the following monotonicity property holds

$$\langle Y' \circ T_\mu^{\mu'} - Y, T_\mu^{\mu'} - I \rangle_\mu \ge 0. \tag{13}$$

If the set $\partial \mathcal{F}(\mu) \subset L^2(\mu; \mathsf{X})$ of subgradients of $\mathcal{F}$ at $\mu$ is not empty, then there exists a unique element of $\partial \mathcal{F}(\mu)$ with minimal norm. This particular subgradient is denoted $\partial^0 \mathcal{F}(\mu)$. However, the set $\partial \mathcal{F}(\mu)$ might be empty. A typical condition for nonemptiness requires the density $\mu(x)$ to have some Sobolev regularity. For every open set $\Omega \subset \mathsf{X}$, we denote $S^{1,1}(\Omega)$ the Sobolev space of Leb-integrable functions $u : \Omega \to \mathbb{R}$ admitting a Leb-integrable weak gradient $\nabla u : \Omega \to \mathsf{X}$. We say that $u \in S_{\mathrm{loc}}^{1,1}(\Omega)$ if $u \in S^{1,1}(K)$ for every bounded open set $K \subset \Omega$. Obviously, $S^{1,1}(\Omega) \subset S_{\mathrm{loc}}^{1,1}(\Omega)$.

## 2.3 Assumptions on $F$ and $G$

Consider $F : \mathsf{X} \to \mathbb{R}$ and $G : \mathsf{X} \to (-\infty, +\infty]$. We make the following assumptions.

**Assumption 1.** The function $F$ is convex and $L$-smooth. Moreover, $G \in \Gamma_0(\mathsf{X})$.

Note that $V := F + G \in \Gamma_0(\mathsf{X})$. We denote $\lambda_F$ (resp. $\lambda_{G^*}$) the strong convexity parameter of $F$ (resp. $G^*$), equal to zero if $F$ (resp. $G^*$) is not strongly convex.

**Assumption 2.** The integral $\int \exp(-V)d\,\mathrm{Leb}$ is positive and finite.

Assumption 2 is needed to define the target distribution $\mu^\star \propto \exp(-V)$, and implies that $\mathrm{int}(D) \neq \emptyset$, where $D := \mathrm{dom}(V)$.

**Lemma 1.** If Assumptions 1 and 2 hold, then

$$\int |V(x)| \exp(-V(x))dx < \infty, \quad \text{and} \quad \int \|x\|^2 \exp(-V(x))dx < \infty.$$

Lemma 1 implies that $\mu^\star \in \mathcal{P}_2(\mathsf{X})$ and using Assumption 1, $\|\nabla F\| \in L^2(\mu^\star; \mathbb{R})$. Since $G \in \Gamma_0(\mathsf{X})$, $G$ is differentiable Leb-a.e. (almost everywhere) on $\mathrm{int}(D)$, see [24, Theorem 25.5].

**Assumption 3.** The integral $\int_{\mathrm{int}(D)} \|\nabla G\|^2 \exp(-V)d\,\mathrm{Leb}$ is finite.

Assumption 3 is equivalent to requiring $\|\nabla G\| \in L^2(\mu^\star; \mathbb{R})$, see below. Moreover, we assume the following regularity property for the function $\exp(-V)$.

**Assumption 4.** The function $\exp(-V)$ belongs to the space $S^{1,1}_{\mathrm{loc}}(\mathsf{X})$.

Assumption 4 is a necessary condition for $\partial \mathcal{H}(\mu^\star) \neq \emptyset$, see below. This assumption precludes $\mu^\star$ from being a uniform distribution. However, Assumption 4 is quite general, *e.g.*, $\exp(-V)$ need not be continuous or positive (see the numerical experiment section). Finally, we assume that the stochastic gradients of $F$ have a bounded variance. Consider an abstract measurable space $(\Xi, \mathscr{G})$, and a random variable $\xi$ with values in $(\Xi, \mathscr{G})$.

**Assumption 5.** For every $x \in \mathsf{X}$, $f(x, \xi)$ is integrable and $F(x) = \mathbb{E}_\xi(f(x, \xi))$. Moreover, there exists $\sigma_F \geq 0$ such that for every $x \in \mathsf{X}$, $\mathbb{V}_\xi(\|\nabla f(x, \xi)\|) \leq \sigma_F^2$, where $\mathbb{V}$ denotes the variance.

The last assumption implies that the stochastic gradients are unbiased: $\mathbb{E}_\xi(\nabla f(x, \xi)) = \nabla F(x)$ for every $x \in \mathsf{X}$.

## 3 Primal dual optimality in Wasserstein space

Let $\mathcal{F} : \mathcal{P}_2(\mathsf{X}) \to (-\infty, +\infty]$ be defined by

$$\mathcal{F}(\mu) := \mathcal{H}(\mu) + \mathcal{E}_V(\mu) = \mathcal{H}(\mu) + \mathcal{E}_F(\mu) + \mathcal{E}_G(\mu). \tag{14}$$

Using Lemma 1, $\mathcal{H}(\mu^\star)$ and $\mathcal{E}_V(\mu^\star)$ are finite real numbers. Moreover, using [19, Lemma 1.b], for every $\mu \in \mathcal{P}_2(\mathsf{X})$ such that $\mathcal{E}_V(\mu) < \infty$, we have the identity

$$\mathcal{F}(\mu) - \mathcal{F}(\mu^\star) = \mathrm{KL}(\mu | \mu^\star). \tag{15}$$

Equation (15) says that $\mu^\star$ is the unique minimizer of $\mathcal{F}$: $\mu^\star = \arg\min \mathcal{F}$.

### 3.1 Subdifferential calculus

The following result is a consequence of [1, Theorem 10.4.13].

**Theorem 2.** Let $\mu \propto \rho$ be an element of $\mathrm{dom}(\mathcal{F})$. Then, $\mathrm{supp}(\mu) \subset \overline{D}$ and $\mu(\overline{D} \setminus \mathrm{int}(D)) = 0$. Moreover, $\partial \mathcal{F}(\mu) \neq \emptyset$ if and only if $\rho \in S^{1,1}_{\mathrm{loc}}(\mathrm{int}(D))$ and there exists $w \in L^2(\mu)$ such that

$$w(x)\rho(x) = \nabla \rho(x) + \rho(x)\nabla V(x), \tag{16}$$

for $\mu$-a.e. $x$. In this case, $w = \partial^0 \mathcal{F}(\mu)$.

If Assumptions 1 and 2 hold, then $\mathcal{F}(\mu^\star) < \infty$ using Lemma 1. Then, Theorem 2 implies that $\mu^\star(\text{int}(D)) = 1$. Therefore, using [24, Theorem 25.5], $G$ and $V$ are $\mu^\star$-a.s. differentiable.

Moreover, applying Theorem 2 with $V \equiv 0$, we can replace $\mathcal{F}$ by $\mathcal{H}$ and $D$ by $\mathsf{X}$. We obtain that $\boldsymbol{\partial}\mathcal{H}(\mu) \neq \emptyset$ if and only if $\rho \in S^{1,1}_{\text{loc}}(\mathsf{X})$ and $w\rho = \nabla\rho$ for some $w \in L^2(\mu; \mathsf{X})$. Now, we set $\mu = \mu^\star$ and $\rho = \exp(-V)$. Using Assumption 4 and $w = -\nabla V$, we obtain that $\boldsymbol{\partial}^0\mathcal{H}(\mu^\star) = -\nabla V$ $\mu^\star$-a.e. Therefore, using that $\nabla G$ is well defined $\mu^\star$-a.e., $\mu^\star$ satisfies

$$0 = \nabla F(x) + \boldsymbol{\partial}^0\mathcal{H}(\mu^\star)(x) + \nabla G(x), \text{ for } \mu^\star - \text{a.e. } x. \tag{17}$$

Equation (17) can be seen as the first order optimality conditions associated with the minimization of the functional $\mathcal{F}$. Consider the "dual" variable $Y^\star : x \mapsto \nabla G(x)$ defined $\mu^\star$ a.e. Using Assumption 3 and $\mu^\star(\text{int}(D)) = 1$, $Y^\star \in L^2(\mu^\star; \mathsf{X})$. We can express the first order optimality condition (17) as $0 = \nabla F(x) + \boldsymbol{\partial}^0\mathcal{H}(\mu^\star)(x) + Y^\star(x)$, $\mu^\star$ a.e. Besides, $Y^\star(x) \in \partial G(x)$, therefore $0 \in -x + \partial G^*(Y^\star(x))$ using $\partial G^* = (\partial G)^{-1}$. Denote $\nu^\star := Y^\star \# \mu^\star \in \mathcal{P}_2(\mathsf{X})$ and $\pi^\star := (I, Y^\star)\#\mu^\star \in \mathcal{P}_2(\mathsf{X}^2)$. The relationship between $\mu^\star$ and $Y^\star$ can be summarized as

$$\begin{bmatrix} 0 \\ 0 \end{bmatrix} \in \begin{bmatrix} \nabla F(x) + \boldsymbol{\partial}^0\mathcal{H}(\mu^\star)(x) & +y \\ -x & +\partial G^*(y) \end{bmatrix} \text{ for } \pi^\star \text{ a.e. } (x, y). \tag{18}$$

In the sequel, we fix the probability space $(\Omega, \mathscr{F}, \mathbb{P}) = (\mathsf{X}^2, \mathcal{B}(\mathsf{X}^2), \pi^\star)$, denote $\mathbb{E}$ the mathematical expectation and $L^2$ the space $L^2(\Omega, \mathscr{F}, \mathbb{P}; \mathsf{X})$. The expression "almost surely" (a.s.) will be understood w.r.t. $\mathbb{P}$. Recall that $x^\star$ is the map $(x, y) \mapsto x$ and $y^\star : (x, y) \mapsto y$. Using Assumption 3, $x^\star, y^\star \in L^2$, $x^\star \sim \mu^\star$, $y^\star \sim \nu^\star$, $(x^\star, y^\star) \sim \pi^\star$ and $y^\star = \nabla G(x^\star)$ a.s.

### 3.2 Lagrangian function and duality gap

We introduce the following Lagrangian function defined for every $\mu \in \mathcal{P}_2(\mathsf{X})$ and $y \in L^2$ by

$$\mathscr{L}(\mu, y) := \mathcal{E}_F(\mu) + \mathcal{H}(\mu) - \mathcal{E}_{G^*}(\nu) + \mathbb{E}\langle x, y\rangle, \tag{19}$$

where $x = T^\mu_{\mu^\star}(x^\star)$. This Lagrangian is similar to the one used in Euclidean optimization; see the appendix. We also define the duality gap by

$$\mathscr{D}(\mu, y) := \mathscr{L}(\mu, y^\star) - \mathscr{L}(\mu^\star, y). \tag{20}$$

The next theorem, which is of independent interest, can be interpreted as a strong duality result for the Lagrangian function $\mathscr{L}$, see [24, Lemma 36.2].

**Theorem 3** (Strong duality). Let Assumptions 1–4 hold true. Then, for every $\mu \in \mathcal{P}_2(\mathsf{X}), y \in L^2$, $\mathscr{D}(\mu, y) \geq 0$ and $\mathscr{L}(\mu, y) \leq \mathcal{F}(\mu)$. Moreover, $(\mu^\star, y^\star)$ is a saddle point of $\mathscr{L}$ with saddle value $\mathcal{F}(\mu^\star)$, i.e.,

$$\mathscr{L}(\mu^\star, y) \leq \mathcal{F}(\mu^\star) = \mathscr{L}(\mu^\star, y^\star) \leq \mathscr{L}(\mu, y^\star). \tag{21}$$

Finally, $\mathscr{L}(\mu^\star, y) = \mathcal{F}(\mu^\star)$ if and only if $y = y^\star$, and, if $F$ is strictly convex, $\mathcal{F}(\mu^\star) = \mathscr{L}(\mu, y^\star)$ if and only if $\mu = \mu^\star$.

The proof of Theorem 3 relies on using (18) to write the duality gap as the sum of the Bregman divergences of $F$, $G^*$ and $\mathcal{H}$. We shall use the nonnegativity of the duality gap to derive convergence bounds for PSGLA.

## 4 Forward Backward representation of PSGLA

In this section, we present our viewpoint on PSGLA (3). More precisely, we represent PSGLA as a (stochastic) Forward Backward algorithm involving (stochastic) monotone operators which are not necessarily subdifferentials.

**Intuition.** Let $\pi \in \mathcal{P}_2(\mathsf{X}^2)$ and consider $A, B(\pi) \in L^2(\pi; \mathsf{X}^2)$ the set valued maps

$$A : (x, y) \mapsto \begin{bmatrix} y \\ -x & +\partial G^*(y) \end{bmatrix}, \quad B(\pi) : (x, y) \mapsto \begin{bmatrix} \nabla F(x) + \boldsymbol{\partial}\mathcal{H}(\mu)(x) \\ 0 \end{bmatrix}, \tag{22}$$

where $\mu = x^\star\#\pi$. The maps $\pi \mapsto A$ and $\pi \mapsto B(\pi)$ satisfy a monotonicity property similar to (13) (note that $A$ is a maximal monotone operator as the sum of $S : (x, y) \mapsto (y, -x)$ and the subdifferential of the $\Gamma_0(\mathsf{X}^2)$ function $(x, y) \mapsto G^*(y)$). Inclusion (18) can be rewritten as

$$0 \in (A + B(\pi^\star))(x, y), \text{ for } \pi^\star \text{ a.e. } (x, y). \tag{23}$$

**Rigorous Forward Backward representation.** The "monotone" inclusion (23) intuitively suggests the following stochastic Forward Backward algorithm for obtaining samples from $\pi^\star$ (and hence from $\mu^\star$ by marginalizing):

$$P \begin{bmatrix} x^{k+1/2} - x^k \\ y^{k+1/2} - y^k \end{bmatrix} = -\gamma \begin{bmatrix} \nabla f(x^k, \xi^{k+1}) - \sqrt{\frac{2}{\gamma}} W^{k+1} \\ 0 \end{bmatrix} \tag{24}$$

$$P \begin{bmatrix} x^{k+1} - x^{k+1/2} \\ y^{k+1} - y^{k+1/2} \end{bmatrix} \in -\gamma A(x^{k+1}, y^{k+1}). \tag{25}$$

Above, $P \in \mathbb{R}_{++}^{d \times d}$ is an appropriately chosen matrix. Indeed, Algorithm (24)-(25) looks like a stochastic Forward-Backward algorithm [6, 7, 15, 26] where the gradient is perturbed by a Gaussian vector, as in the Langevin algorithm (1). In Algorithm (24)-(25), we cannot set $P$ to be the identity map of $\mathsf{X}^2$ because the inclusion (25) is intractable in this case. We take $P : (x, y) \mapsto x$, *i.e.*, with our notations, $P = x^\star$. Although the matrix $P$ is only semi-definite positive, the next lemma shows that Algorithm (24)-(25) is still well defined. More precisely, the next lemma shows that $x^{k+1} = \mathrm{prox}_{\gamma G}(x^{k+1/2})$ (by taking $z = (x^{k+1/2}, y^{k+1/2})$ in the lemma) and hence the resulting algorithm (24)-(25) is PSGLA. Based on the representation (24)-(25) of PSGLA, the next lemma also provides an important inequality used later in the proof of Theorem 5.

**Lemma 4.** Let $z = (x, y), z' = (x', y') \in \mathsf{X}^2$. Then $P(z' - z) \in -\gamma A(z')$ if and only if $x' = \mathrm{prox}_{\gamma G}(x)$ and $y' = \mathrm{prox}_{G^*/\gamma}(x/\gamma)$. Moreover, if $G \in \Gamma_0(\mathsf{X})$ is $1/\lambda_{G^*}$-smooth, then

$$\|x' - x^\star\|^2 \leq \|x - x^\star\|^2 - 2\gamma \left( G^*(y') - G^*(y^\star) - \langle y', x^\star \rangle + \langle y^\star, x \rangle \right)$$
$$- \gamma(\lambda_{G^*} + \gamma)\|y' - y^\star\|^2 + \gamma^2 \|y^\star\|^2. \tag{26}$$

## 5  Main results

We now provide our main result on PSGLA (3). For $r \in \mathbb{N}/2$, denote $\mu^r$ (resp. $\nu^r$) the distribution of $x^r$ (resp. $y^r$), defined in the previous section.

**Theorem 5.** Let Assumptions 1, 2, 3 and 5 hold true. If $F$ is $\lambda_F$-strongly convex and $G$ is $1/\lambda_{G^*}$-smooth, then for every $\gamma \leq 1/L$,

$$W^2(\mu^{k+1}, \mu^\star) \leq (1 - \gamma \lambda_F) W^2(\mu^k, \mu^\star) - \gamma(\lambda_{G^*} + \gamma) W^2(\nu^{k+1}, \nu^\star)$$
$$- 2\gamma \left( \mathscr{L}(\mu^{k+1/2}, y^\star) - \mathscr{L}(\mu^\star, y_\star^{k+1}) \right) + \gamma^2 C, \tag{27}$$

where $C := \int_{\mathrm{int}(D)} \|\nabla G(x)\|^2 d\mu^\star(x) + 2(Ld + \sigma_F^2)$ and $y_\star^{k+1} := \mathrm{prox}_{G^*/\gamma}(x_\star^{k+1/2}/\gamma) \sim \nu^{k+1}$, where $x_\star^{k+1/2} := T_{\mu^\star}^{\mu^{k+1/2}}(x^\star)$.

The proof of Theorem 5 relies on using Lemma 4 along with [19, Lemma 30]. Inspecting the proof of Theorem 5, one can see that any $\bar{\mu}, \bar{y}$ can replace $\mu^\star, y^{\star 5}$. The situation is similar to primal dual algorithms in optimization [11, 18] and Evolution Variational Inequalities in optimal transport [1].

The next corollary is obtained by using $\mathscr{D}(\mu^{k+1/2}, y_\star^{k+1}) \geq 0$ (Theorem 3) and iterating (27).

**Corollary 6.** Let Assumptions 1–5 hold true. If $\gamma \leq 1/L$, then

$$\min_{j \in \{0, \ldots, k-1\}} \mathscr{D}(\mu^{j+1/2}, y_\star^{j+1}) \leq \frac{1}{2\gamma k} W^2(\mu^0, \mu^\star) + \frac{\gamma}{2} C, \tag{28}$$

$$\min_{j \in \{1, \ldots, k\}} W^2(\nu^j, \nu^\star) \leq \frac{1}{\gamma(\lambda_{G^*} + \gamma)k} W^2(\mu^0, \mu^\star) + \frac{\gamma}{\lambda_{G^*} + \gamma} C. \tag{29}$$

Finally, if $\lambda_F > 0$, then

$$W^2(\mu^k, \mu^\star) \leq (1 - \gamma \lambda_F)^k W^2(\mu^0, \mu^\star) + \frac{\gamma}{\lambda_F} C. \tag{30}$$

If $G$ is Lipschitz continuous (in particular if $G \equiv 0$), then our Assumptions hold true. Moreover, inequality (28) recovers [19, Corollary 18] but with the duality gap instead of the KL

divergence. Obtaining a result in terms of KL divergence is hopeless for PSGLA in general because the KL divergence is infinite; see the appendix. Connecting the convergence of the duality gap to zero to known modes of convergence is left for future work. Besides, obtaining an inequality like (29) that holds when $F$ is just convex is rather not standard in the literature on Langevin algorithm, see [25, 35]. Corollary 6 implies the following complexity results. Given $\varepsilon > 0$, choosing $\gamma = \min(1/L, \varepsilon/C)$ and $k \geq \max(L/\varepsilon, C/\varepsilon^2)W^2(\mu^0, \mu^\star)$ in inequality (28) leads to $\min_{j \in \{0,\dots,k-1\}} \mathscr{D}(\mu^{j+1/2}, y_\star^{j+1}) \leq \varepsilon$. If $\lambda_{G^*} > 0$ (*i.e.*, if $G$ is smooth), choosing $\gamma = \min(1/L, \frac{\lambda_{G^*}\varepsilon}{2C})$ and $k \geq \max(\frac{2L}{\lambda_{G^*}\varepsilon}, \frac{4C}{\lambda_{G^*}^2 \varepsilon^2})W^2(\mu^0, \mu^\star)$ in inequality (29) leads to $\min_{j \in \{1,\dots,k\}} W^2(\nu^j, \nu^\star) \leq \varepsilon$. Finally, if $\lambda_F > 0$ (*i.e.*, if $F$ is strongly convex), choosing $\gamma = \min(1/L, \frac{\lambda_F\varepsilon}{2C})$ and $k \geq \frac{1}{\gamma \lambda_F}\log(2W^2(\mu^0,\mu^\star)/\varepsilon)$ *i.e.*,

$$k \geq \max\left(\frac{L}{\lambda_F}, \frac{2C}{\lambda_F^2 \varepsilon}\right) \log\left(\frac{2W^2(\mu^0,\mu^\star)}{\varepsilon}\right), \quad C = \int_{\text{int}(D)} \|\nabla G(x)\|^2 d\mu^\star(x) + 2(Ld + \sigma_F^2) \quad (31)$$

in inequality (30), leads to $W^2(\mu^k, \mu^\star) \leq \varepsilon$. [6] In the case where $G$ is $M$-Lipschitz continuous, the complexity (31) improves [19, Corollary 22] since $\int_{\text{int}(D)} \|\nabla G(x)\|^2 d\mu^\star(x) \leq M^2$ .

## 6 Conclusion

We made a step towards theoretical understanding the properties of the Langevin algorithm in the case where the target distribution is not smooth and not fully supported. This case is known to be difficult to analyze and has many applications [9, 10, 21]. Our analysis improves and extends the state of the art.

Moreover, our approach is new. We developed a primal dual theory for a minimization problem over the Wasserstein space, which is of independent interest. A broader duality theory for minimization problems in the Wasserstein space would be of practical and theoretical interest.

## 7 Acknowledgement

We thank Laurent Condat for introducing us to the primal dual view of the proximal gradient algorithm in Hilbert spaces.

## 8 Broader impact

Our work contributes to the understanding of a sampling algorithm used in statistics. Our main results are of theoretical nature (convergence rates). Therefore, we do not see any immediate societal impact of our results.

## Footnotes

[1] A convex body is a compact convex set with a nonempty interior.

[2] Our big O notation ignores logarithm factors.

[3]This result also holds if $F$ is not strongly convex.

[4]The operators $P^{-1}A$ and $P^{-1}B$ are not monotone in general, however they are monotone under the inner product induced by $P$.

[5] The proof does not rely on specific properties of the latter like being primal dual optimal.

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
