[Supplementary Material]

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

# Appendix

## Contents

# A Numerical experiments

In this section, we illustrate our results on PSGLA through numerical experiments.

**Sampling *a posteriori*.** We consider a statistical framework where i.i.d. random vectors (data) $D_1, \dots, D_n$ with distribution $\mathbb{P}_{x^\star}$ are observed. We adopt a Bayesian strategy where we assume the distribution $\mathbb{P}_{x^\star}$ to be indexed by a random vector $x^\star$ with values in X. Denote $\mathcal{L}(\cdot, x^\star)$ the density of $\mathbb{P}_{x^\star}$ (a.k.a. the likelihood function) w.r.t. some reference measure. Given a prior distribution for $x^\star$ with density $\pi$ w.r.t. Leb, our goal is construct samples $x^1, \dots, x^k$ from the posterior distribution

$$\mu^\star(x | D_1, \dots, D_n) \propto \pi(x) \prod_{i=1}^{n} \mathcal{L}(D_i, x), \tag{32}$$

in order *e.g.* to estimate the mean *a posteriori* via Monte Carlo approximations,

$$m^\star := \int x \mu^\star(x | D_1, \dots, D_n) d \operatorname{Leb}(x) \simeq \frac{1}{k} \sum_{j=1}^{k} x^j. \tag{33}$$

**Wishart distribution.** In the experiments, the Euclidean space X is a space of $d \times d$ symmetric matrices and $\pi$ is the Wishart distribution defined by

$$\pi(x) \propto |\det(x)|^{\frac{\nu - d - 1}{2}} \exp\left(-\frac{\operatorname{tr}(V^{-1} x)}{2}\right) \mathbf{1}_{\mathbb{R}_{++}^{d \times d}}(x), \tag{34}$$

where $\nu > d - 1$ and $V \in \mathbb{R}_{++}^{d \times d}$ are parameters of the distribution. Note that $\pi(x) = 0$ if $x$ is not a positive definite matrix. The mean of the Wishart distribution is equal to $\nu V$. The Wishart distribution is widely used in Random matrix theory and applications, see [29]. Indeed, the Wishart distribution is a conjugate prior to the Gaussian likelihood. More precisely, assume that for every $D \in \mathbb{R}^d$ and $x \in \mathbb{R}_{++}^{d \times d}$,

$$\mathcal{L}(D, x) = \frac{1}{\sqrt{2\pi}^d} \exp\left(-\frac{1}{2} D^T x D\right) \sqrt{\det(x)}, \tag{35}$$

is the density of a centered Gaussian distribution with precision matrix (*i.e.*, inverse variance-covariance matrix) $x$. Then, if $\pi$ is Wishart with parameters $\nu$ and $V$ (*i.e.*, $\pi$ is given by (34)), then the posterior distribution $\mu^\star(\cdot | D_1, \dots, D_n)$ (32) is Wishart with parameters $\nu' = n + \nu$ and $V' = \left(I + \sum_{i=1}^{n} D_i D_i^T\right)^{-1}$:

$$\mu^\star(x | D_1, \dots, D_n) \propto |\det(x)|^{\frac{(\nu + n) - d - 1}{2}} \exp\left(-\frac{\operatorname{tr}\left((V^{-1} + \sum_{i=1}^{n} D_i D_i^T) x\right)}{2}\right) \mathbf{1}_{\mathbb{R}_{++}^{d \times d}}(x). \tag{36}$$

Moreover, the mean of the posterior distribution is equal to

$$m^\star = (n + \nu) \left(I + \sum_{i=1}^{n} D_i D_i^T\right)^{-1}. \tag{37}$$

**Setup.** We consider two *a posterori* sampling problems.

First, we consider the task of learning the mean of the data. More precisely, $\mathbb{P}_{x^\star}$ is a Gaussian distribution over $\mathbb{R}$ with mean $x^\star$ and unit variance. We use the Wishart distribution $\pi$ with $V = I$ as the prior distribution. Note that in this one dimensional case, the Wishart distribution $\pi$ boils down to a Gamma distribution over $\mathbb{R}$.

In other words, $\pi(x) \propto \exp(-G(x))$ and $\mu^\star(x | D_1, \dots, D_n) \propto \exp(-G(x) - \sum_{i=1}^{n} f_i(x))$ where the functions $f_i, G \in \Gamma_0(\mathsf{X})$ are defined by

$$G(x) := -\frac{\nu - d - 1}{2} \log |x| + \frac{x}{2} + \iota_{(0, +\infty)}(x),$$

$$f_i(x) := \frac{|x - D_i|^2}{2},$$

for every $i \in \{1, \dots, n\}$. The data points $D_i$ are generated randomly using a Gaussian distribution. Note that $f_i$ is smooth and strongly convex. Moreover, $G$ is nonsmooth and the proximity operator of $G$ has a closed form thanks to recent results.[7] We consider $d = 1$ in order to be able to represent the numerical results with histograms, see Figures 1-2.

Figure 1: Histograms drawn by the $k$ iterates of PSGLA and MYULA($\lambda$), for various values of $\lambda$, compared to the target distribution $\mu^\star$. Case $d = 1, \gamma = 1.0$.

Figure 2: Histograms drawn by the $k$ iterates of PSGLA and MYULA($\lambda$), for various values of $\lambda$, compared to the target distribution $\mu^\star$. Case $d = 1, \gamma = 0.1$.

Then, we consider the task of learning the precision matrix of the data. More precisely, $\mathbb{P}_{x^\star}$ is a centered Gaussian distribution over $\mathbb{R}^d$ with precision matrix $x^\star$. We use the Wishart distribution $\pi$ with $V = I$ as the prior distribution. Since the prior distribution $\pi$ (34) is conjugate to the likelihood function $\mathcal{L}(\cdot, x^\star)$ (35), the posterior distribution $\mu^\star$ is given by (36). Thus, we can use $\mu^\star$ as a ground

truth. In other words, $\mu^\star(x|D_1, \ldots, D_n) \propto \exp(-G(x) - F(x))$ where the functions $F, G \in \Gamma_0(\mathsf{X})$ are defined by

$$G(x) := -\frac{(\nu + n) - d - 1}{2} \log|\det(x)| + \frac{\operatorname{tr}(x)}{2} + \iota_{\mathbb{R}_{++}^{d \times d}}(x),$$

$$F(x) := \sum_{i=1}^{n} \frac{\operatorname{tr}(D_i D_i^T x)}{2}.$$

The data points $D_i$ are generated randomly using a Gaussian distribution. Note that $F$ is smooth and convex, hence Assumptions 1-2 are satisfied. Moreover, Assumptions 3-4 are satisfied *e.g.* if $n + \nu > d + 3$. Finally, $G$ is nonsmooth and the proximity operator of $G$ has a closed form [4, Corollary 24.65]. We consider several values of $d$: $d = 1, d = 10$ and $d = 100$. The number of entries of the iterates is $d^2$ and, since the matrices are symmetric, the dimension of the sampling problem is slightly larger than $d^2/2$. Since the mean of $\mu^\star$ is known, we use it as a ground truth and perform a mean *a posteriori* estimation using the estimators (33) constructed by PSGLA and MYULA. The convergence of the estimators is illustrated in Figures 3 and 4 for the cases $d = 10$ and $d = 100$. Moreover, in order to visualize better the multidimensional results of Figures 3 and 4, we consider the *same* sampling problem with $d = 1$ and plot histograms to represent the results, see Figure 5.

Figure 3: Frobenius distance between $m^\star$ and the mean *a posteriori* estimators (ergodic means) constructed by PSGLA and MYULA($\lambda$), for various values of $\lambda, \gamma$ as a function of $k$ in the case $d = 10$.

**Algorithms.** We compare PSGLA to various versions of MYULA [21], parametrized by the smoothing parameter $\lambda > 0$. We use the same learning rate $\gamma$ for the algorithms[8]. We denote these algorithms MYULA($\lambda$). Both MYULA and PSGLA compute one proximity operator $\operatorname{prox}_{\gamma G}$ and

Figure 4: Frobenius distance between $m^\star$ and the mean *a posteriori* estimators (ergodic means) constructed by PSGLA and MYULA($\lambda$), for various values of $\lambda, \gamma$ as a function of $k$ in the case $d = 100$.

one gradient per iteration[9]. They both require to sample one Gaussian random variable over the space of symmetric matrices at each iteration.

**Observations.** Using Langevin algorithm to sample from distributions which are not fully supported is known to be a difficult task [9, 10, 21]. In Figures 1-2 and 5, we see that the shape of the histograms drawn by PSGLA are closer to the target distribution than the shape of the benchmarks histograms. This means that PSGLA converges faster than the benchmarks for this sampling task. This behavior was expected: PSGLA does not introduce extra bias by introducing a smoothing parameter $\lambda$.

*More importantly, we see that the iterates of PSGLA are always feasible i.e., they lie in the support of the target distribution, contrary to the benchmarks.*

Finally, PSGLA is a proximal method, whereas the benchmarks are instances of the standard Langevin algorithm (applied to the smoothed problem depending on $\lambda$). In stochastic optimization, proximal methods are known to be more stable than gradient methods [30]. This phenomenon is observed here. For instance, the range of step sizes allowed by the benchmarks is controlled by the smoothing parameter $\lambda$, see [21]. Using a step size too large for the benchmarks leads to a numerical instability that does not occur for PSGLA.

Figures 3 and 4 are multidimensional extensions of Figure 5. Each figure in 3 and 4 corresponds to a new run. We plotted the convergence of the mean *a posteriori* estimators (*i.e.*, ergodic means) constructed by MYULA($\lambda$) and PSGLA.

Figure 5: Histograms drawn by the $k$ iterates of PSGLA and MYULA($\lambda$), for various values of $\lambda$, compared to the target distribution $\mu^\star$. Case $d = 1, \gamma = 1.0$.

*In general, we see that PSGLA is as good as the MYULA($\lambda$) for the best value of $\lambda$, while conserving the feasibility of the iterates, and without having to select the value of $\lambda$.*

## B    Postponed proofs

### B.1    Proof of Lemma 1

Using [8, Lemma 2.2.1], there exist $A, B > 0$, $\exp(-V(x)) \leq A\exp(-B\|x\|) \leq A$. The last inequality implies $\int \|x\|^2 \exp(-V(x))dx < \infty$. Moreover, $-V(x) \leq \log(A) \leq C :=$ $\max(0, \log(A))$. Using that $u \mapsto u\exp(-u)$ is nonincreasing on $[1, +\infty)$.

$$V(x)\exp(-V(x)) = V(x)\exp(-V(x))\mathbf{1}_{V(x) \leq \|x\|^2+1} + V(x)\exp(-V(x))\mathbf{1}_{V(x) > \|x\|^2+1}$$
$$\leq \left(\|x\|^2 + 1\right)\exp(-V(x)) + \left(\|x\|^2 + 1\right)\exp\left(-(\|x\|^2 + 1)\right).$$

Using $|V(x)| = V(x)\mathbf{1}_{V(x) \geq 0} - V(x)\mathbf{1}_{V(x) < 0} \leq C + V(x)$,

$$|V(x)|\exp(-V(x)) \leq \left(\|x\|^2 + 1 + C\right)A\exp(-B\|x\|) + \left(\|x\|^2 + 1\right)\exp\left(-(\|x\|^2 + 1)\right).$$

We conclude using that the r.h.s. is integrable.

### B.2    Proof of Theorem 3

The proof is divided in six parts, each part proving one claim. Denote $x = T_{\mu^\star}^\mu(x^\star)$.

**Part I.** First,

$$\mathscr{L}(\mu, y^\star) = \mathcal{E}_F(\mu) + \mathcal{H}(\mu) - \mathcal{E}_{G^*}(\nu^\star) + \mathbb{E}\langle x, y^\star\rangle, \tag{38}$$

and

$$\mathscr{L}(\mu^\star, y) = \mathcal{E}_F(\mu^\star) + \mathcal{H}(\mu^\star) - \mathcal{E}_{G^*}(\nu) + \mathbb{E}\langle x^\star, y\rangle. \tag{39}$$

Therefore, the duality gap can be rewritten

$$
\begin{aligned}
\mathscr{D}(\mu, y) =& \mathcal{E}_F(\mu) - \mathcal{E}_F(\mu^\star) + \mathcal{H}(\mu) - \mathcal{H}(\mu^\star) + \mathcal{E}_{G^*}(\nu) - \mathcal{E}_{G^*}(\nu^\star) \\
& + \mathbb{E}\langle x, y^\star\rangle - \mathbb{E}\langle x^\star, y\rangle \\
=& \mathcal{E}_F(\mu) - \mathcal{E}_F(\mu^\star) + \mathcal{H}(\mu) - \mathcal{H}(\mu^\star) + \mathcal{E}_{G^*}(\nu) - \mathcal{E}_{G^*}(\nu^\star) \\
& + \mathbb{E}\langle x - x^\star, y^\star\rangle - \mathbb{E}\langle x^\star, y - y^\star\rangle.
\end{aligned}
$$

Using (18), $y^\star = -\nabla F(x^\star) - \boldsymbol{\partial}^0 \mathcal{H}(\mu^\star)(x^\star)$ and $x^\star \in \partial G^*(y^\star)$.

$$
\begin{aligned}
\mathscr{D}(\mu, y) =& \mathcal{E}_F(\mu) - \mathcal{E}_F(\mu^\star) - \mathbb{E}\langle\nabla F(x^\star), x - x^\star\rangle \\
& + \mathcal{H}(\mu) - \mathcal{H}(\mu^\star) - \mathbb{E}\langle\boldsymbol{\partial}^0\mathcal{H}(\mu^\star)(x^\star), x - x^\star\rangle \\
& + \mathcal{E}_{G^*}(\nu) - \mathcal{E}_{G^*}(\nu^\star) - \mathbb{E}\langle x^\star, y - y^\star\rangle \\
=& \mathbb{E}F(x) - \mathbb{E}F(x^\star) - \mathbb{E}\langle\nabla F(x^\star), x - x^\star\rangle \\
& + \mathcal{H}(\mu) - \mathcal{H}(\mu^\star) - \langle\boldsymbol{\partial}^0\mathcal{H}(\mu^\star), T^\mu_{\mu^\star} - I\rangle_{\mu^\star} \\
& + \mathbb{E}G^*(y) - \mathbb{E}G^*(y^\star) - \mathbb{E}\langle x^\star, y - y^\star\rangle,
\end{aligned}
\tag{40}
$$

where the last equality comes from the transfer theorem. We get $\mathscr{D}(\mu, y) \geq 0$ using the convexity of $F$, the convexity of $G^*$ and the geodesic convexity of $\mathcal{H}$ (inequality (12)).

**Part II.** Since $\mathscr{D}(\mu, y^\star) \geq 0$ and $\mathscr{D}(\mu^\star, y) \geq 0$,

$$\mathscr{L}(\mu^\star, y) \leq \mathscr{L}(\mu^\star, y^\star) \leq \mathscr{L}(\mu, y^\star). \tag{41}$$

**Part III.** Then,

$$
\begin{aligned}
\mathscr{L}(\mu, y) &= \mathcal{E}_F(\mu) + \mathcal{H}(\mu) + \mathbb{E}\left(\langle x, y\rangle - G^*(y)\right) \tag{42} \\
&\leq \mathcal{E}_F(\mu) + \mathcal{H}(\mu) + \mathbb{E}\left(\sup\langle x, \cdot\rangle - G^*\right) \tag{43} \\
&\leq \mathcal{E}_F(\mu) + \mathcal{H}(\mu) + \mathbb{E}G(x) \tag{44} \\
&= \mathcal{F}(\mu),
\end{aligned}
$$

using [4, Proposition 13.15].

**Part IV.** Using [4, Proposition 16.10], $\sup\langle x, \cdot\rangle - G^* = G(x)$, and $\langle x, y\rangle - G^*(y) = G(x)$ if and only if $y \in \partial G(x)$ (or $x \in \partial G^*(y)$). Taking $\mu = \mu^\star$ and $y = y^\star$ in (42), we have $x^\star = x \in \partial G^*(y^\star)$ and therefore $\mathscr{L}(\mu^\star, y^\star) = \mathcal{F}(\mu^\star)$.

**Part V.** Assume that $\mathscr{L}(\mu^\star, \bar{y}) = \mathcal{F}(\mu^\star)$. We shall prove that $\bar{y} = y^\star$ a.s. Since $\mathcal{F}(\mu^\star) \leq \mathscr{L}(\mu^\star, \bar{y})$, inequality (43) becomes an equality when $\mu = \mu^\star$ and $y = \bar{y}$. Therefore, $\langle x^\star, \bar{y}\rangle - G^*(\bar{y}) = \sup\langle x^\star, \cdot\rangle - G^* = G(x^\star)$ a.s., which implies $\bar{y} \in \partial G(x^\star) = \{\nabla G(x^\star)\} = \{y^\star\}$ a.s.

**Part VI.** Assume that $\mathscr{L}(\mu^\star, y^\star) = \mathscr{L}(\bar{\mu}, y^\star)$ and that $F$ is strictly convex. We shall prove that $\mu^\star = \bar{\mu}$. We have $\mathscr{D}(\bar{\mu}, y^\star) = 0$, and, using (40),

$$\mathscr{D}(\bar{\mu}, y^\star) = \mathbb{E}F(\bar{x}) - \mathbb{E}F(x^\star) - \mathbb{E}\langle\nabla F(x^\star), \bar{x} - x^\star\rangle + \mathcal{H}(\bar{\mu}) - \mathcal{H}(\mu^\star) - \langle\boldsymbol{\partial}^0\mathcal{H}(\mu^\star), T^{\bar{\mu}}_{\mu^\star} - I\rangle_{\mu^\star}, \tag{45}$$

where $\bar{x} = T^{\bar{\mu}}_{\mu^\star}(x^\star)$. Using the convexity of $F$ and the geodesic convexity of $\mathcal{H}$, $F(\bar{x}) - F(x^\star) - \langle\nabla F(x^\star), \bar{x} - x^\star\rangle = 0$ a.s. Using the strict convexity of $F$, $\bar{x} = x^\star$ a.s., therefore $\bar{\mu} = \mu^\star$.

### B.3  Proof of Lemma 4

If $x' = x - \gamma y'$ and $0 \in -x' + \partial G^*(y')$, we have $y' \in \partial G(x')$ and $x' = \mathrm{prox}_{\gamma G}(x)$. Moreover, $0 \in -x + \gamma y' + \partial G^*(y')$ implies $y' = \mathrm{prox}_{G^*/\gamma}(x/\gamma)$. One can easily check that this is an

equivalence. Denote $\| \cdot \|_P$ the semi-norm induced by $P$ on $\mathsf{X}^2$ defined by $\|z\|_P = \|x\|$ for every $z = (x, y) \in \mathsf{X}^2$, and $\langle \cdot, \cdot \rangle_P$ the semi-inner product associated. We have

$$\|z' - z^\star\|_P^2 = \|z - z^\star\|_P^2 + 2\langle z' - z, z' - z^\star \rangle_P - \|z' - z\|_P^2. \tag{46}$$

We now identify the terms. First, $\|z' - z^\star\|_P^2 = \|x' - x^\star\|^2$, $\|z - z^\star\|_P^2 = \|x - x^\star\|^2$ and $\|z' - z\|_P^2 = \|x' - x\|^2 = \gamma^2 \|y'\|^2$. Second, using $P(z' - z) \in -\gamma A(z')$, and the definition of $A$, there exists $g^* \in G^*(y')$ such that

$$\langle z' - z, z' - z^\star \rangle_P = \langle P(z' - z), z' - z^\star \rangle = -\gamma\langle y', x' - x^\star \rangle + \gamma\langle x', y' - y^\star \rangle - \gamma\langle g^*, y' - y^\star \rangle. \tag{47}$$

Hence

$$\langle z' - z, z' - z^\star \rangle_P \leq \gamma\langle y', x^\star \rangle - \gamma\langle x', y^\star \rangle - \gamma\left( G^*(y') - G^*(y^\star) + \frac{\lambda_{G^*}}{2}\|y' - y^\star\|^2 \right), \tag{48}$$

using the strong convexity of $G^*$. Plugging into (46),

$$\|x' - x^\star\|^2 \leq \|x - x^\star\|^2 - 2\gamma\left( G^*(y') - G^*(y^\star) + \frac{\lambda_{G^*}}{2}\|y' - y^\star\|^2 - \langle y', x^\star \rangle + \langle y^\star, x' \rangle \right) - \gamma^2\|y'\|^2. \tag{49}$$

Using $x' = x - \gamma y'$,

$$-2\gamma\langle y^\star, x' \rangle - \gamma^2\|y'\|^2 = -2\gamma\langle y^\star, x \rangle - 2\gamma\langle y^\star, -\gamma y' \rangle - \gamma^2\|y'\|^2 = -2\gamma\langle y^\star, x \rangle - \gamma^2\|y' - y^\star\|^2 + \gamma^2\|y^\star\|^2. \tag{50}$$

Plugging into (49) concludes the proof.

## B.4 Proof of Theorem 5

We first recall a standard inequality of the stochastic gradient Langevin algorithm, see *e.g.* [19, Lemma 30].

**Lemma 7** ( [19]). Let Assumptions 1, 2 and 5 hold true. Then, if $F$ is $\lambda_F$-strongly convex, for every $\gamma \leq 1/L$,

$$W^2(\mu^{k+1/2}, \mu^\star) \leq (1 - \gamma\lambda_F)W^2(\mu^k, \mu^\star) + 2\gamma^2(Ld + \sigma_F^2)$$
$$- 2\gamma\left( \mathcal{E}_F(\mu^{k+1/2}) + \mathcal{H}(\mu^{k+1/2}) - \mathcal{E}_F(\mu^\star) - \mathcal{H}(\mu^\star) \right). \tag{51}$$

We now prove Theorem 5. The main tool for the proof is Lemma 4. Replace $x$ by $x_\star^{k+1/2} \sim \mu^{k+1/2}$ in (26). Then $y' = y_\star^{k+1} \sim \nu^{k+1}$ and $\mathrm{prox}_{\gamma G}(x_\star^{k+1/2}) \sim \mu^{k+1}$. Therefore,

$$W^2(\mu^{k+1}, \mu^\star) \leq \mathbb{E}(\|\mathrm{prox}_{\gamma G}(x_\star^{k+1/2}) - x^\star\|^2), \quad W^2(\nu^{k+1}, \nu^\star) \leq \mathbb{E}(\|y_\star^{k+1} - y^\star\|^2).$$

Consequently, taking expectation in (26) we get

$$W^2(\mu^{k+1}, \mu^\star) \leq W^2(\mu^{k+1/2}, \mu^\star) - \gamma(\lambda_{G^*} + \gamma)W^2(\nu^{k+1}, \nu^\star) + \gamma^2 \int \|y\|^2 d\nu^\star(y)$$
$$- 2\gamma\left( \mathcal{E}_{G^*}(\nu^{k+1}) - \mathcal{E}_G^*(\nu^\star) - \mathbb{E}\langle y_\star^{k+1}, x^\star \rangle + \mathbb{E}\langle y^\star, x_\star^{k+1/2} \rangle \right).$$

Combining with Lemma 7, we get the result.

## B.5 Proof of Corollary 6

From Theorem 5,

$$\gamma(\lambda_{G^*} + \gamma)W^2(\nu^{j+1}, \nu^\star) + 2\gamma\mathscr{D}(\mu^{j+1/2}, y_\star^{j+1}) \leq W^2(\mu^j, \mu^\star) - W^2(\mu^{j+1}, \mu^\star) + \gamma^2 C. \tag{52}$$

Summing over $j \in \{0, \dots, k-1\}$,

$$\gamma(\lambda_{G^*} + \gamma) \sum_{j=0}^{k-1} W^2(\nu^{j+1}, \nu^\star) + 2\gamma \sum_{j=0}^{k-1} \mathscr{D}(\mu^{j+1/2}, y_\star^{j+1}) \tag{53}$$

$$\leq W^2(\mu^0, \mu^\star) - W^2(\mu^k, \mu^\star) + k\gamma^2 C. \tag{54}$$

Therefore,

$$\gamma(\lambda_{G^*} + \gamma)k \min_{j \in \{0,\dots,k-1\}} W^2(\nu^{j+1}, \nu^\star) + 2\gamma k \min_{j \in \{0,\dots,k-1\}} \mathscr{D}(\mu^{j+1/2}, y_\star^{j+1}) \quad (55)$$

$$\leq W^2(\mu^0, \mu^\star) + k\gamma^2 C, \quad (56)$$

which implies

$$\min_{j \in \{0,\dots,k-1\}} \mathscr{D}(\mu^{j+1/2}, y_\star^{j+1}) \leq \frac{1}{2\gamma k} W^2(\mu^0, \mu^\star) + \gamma \frac{C}{2}, \quad (57)$$

and,

$$\min_{j \in \{1,\dots,k\}} W^2(\nu^j, \nu^\star) \leq \frac{1}{\gamma(\lambda_{G^*} + \gamma)k} W^2(\mu^0, \mu^\star) + \frac{\gamma}{\lambda_{G^*} + \gamma} C. \quad (58)$$

Moreover, if $\lambda_F > 0$, Theorem 5 implies

$$W^2(\mu^{k+1}, \mu^\star) \leq (1 - \gamma\lambda_F)W^2(\mu^k, \mu^\star) + \gamma^2 C. \quad (59)$$

Iterating, we obtain

$$W^2(\mu^k, \mu^\star) \leq (1 - \gamma\lambda_F)^k W^2(\mu^0, \mu^\star) + \gamma \frac{C}{\lambda_F}. \quad (60)$$

## C  Further intuition on PSGLA

### C.1  Stochastic gradient descent interpretation of the stochastic gradient Langevin algorithm

As mentionned in the introduction, Langevin algorithm can be interpreted as a gradient descent algorithm in the space $\mathcal{P}_2(\mathsf{X})$ to minimize $\mathrm{KL}(\cdot|\mu^\star)$, see *e.g.* [19]. More precisely, consider the case where $G \equiv 0$ and denote $\mu^k$ the distribution of $x^k$. Then PSGLA boils down to the stochastic gradient Langevin algorithm (*i.e.*, PSGLA without proximal step) and satisfy the following inequality (Lemma 7)

$$W^2(\mu^{k+1}, \mu^\star) \leq (1 - \gamma\lambda_F)W^2(\mu^k, \mu^\star) - 2\gamma\left(\mathcal{F}(\mu^{k+1}) - \mathcal{F}(\mu^\star)\right) + 2\gamma^2(Ld + \sigma_F^2), \quad (61)$$

if $F$ is $L$-smooth, $\lambda_F$-strongly convex and $\gamma \leq 1/L$. The last inequality is similar to a standard inequality used in the analysis of SGD. More precisely, the analysis of SGD often relies on an inequality similar to (61), by replacing the Wasserstein distance by the Euclidean distance and $\mathcal{F}$ by the objective function to be minimized by SGD (note that $Ld + \sigma_F^2$ is a constant). Therefore, unrolling the recursion (61) (which is the standard way to obtain convergence rates for SGD) leads to the complexity $\mathcal{O}(1/\varepsilon^2)$ in terms of objective gap $\mathcal{F}(\mu) - \mathcal{F}(\mu^\star)$. Using (15), recall that the objective gap is the KL divergence.

In this paper, we considered the case $G \neq 0$. One can obtain an inequality similar to (61) for PSGLA if $G$ is Lipschitz continuous, see [19, 27]. However, for a general $G \in \Gamma_0(\mathsf{X})$, it is hopeless. Indeed, $\mathcal{F}(\mu^{k+1}) - \mathcal{F}(\mu^\star) = +\infty$ in general because $\mathcal{H}(\mu^{k+1}) = +\infty$ since $\mu^{k+1}$ is not absolutely continuous w.r.t. Leb (*e.g.* when the proximal step is a projection). Moreover, $\mathcal{F}(\mu^{k+1/2}) - \mathcal{F}(\mu^\star) = +\infty$ in general because $\mathcal{E}_G(\mu^{k+1/2}) = +\infty$ since $\mu^{k+1/2}$ is not supported by $\mathrm{dom}(G)$ ($\mathrm{supp}(\mu^{k+1/2}) = \mathsf{X}$ because of the Gaussian noise). Therefore, one cannot obtain a rate in terms of KL divergence (*i.e.*, objective gap) for PSGLA in general, since the KL divergence is equal to $+\infty$.

On order to overcome this difficulty, we assumed 4 and adopted a primal dual interpretation of PSGLA where PSGLA is seen as a Forward Backward algorithm involving monotone operators. We obtained an inequality similar to (61), but with the duality gap instead of the objective gap, and we proved that the duality gap is nonnegative.

### C.2  Primal dual interpretation of the proximal gradient algorithm

The approach of this paper can also be used to interpret the proximal gradient algorithm as a primal dual algorithm.

Consider the minimization problem

$$\min_{x \in \mathsf{X}} F(x) + G(x). \quad (62)$$

To solve Problem (62), the proximal gradient algorithm is written
$$x^{k+1} = \text{prox}_{\gamma G}\left(x^k - \gamma \nabla F(x^k)\right). \tag{63}$$
The proximal gradient algorithm can be seen as a primal dual algorithm for Problem (62) [24].

Indeed, a solution $x^\star$ to Problem (62) satisfies $0 \in \nabla F(x^\star) + \partial G(x^\star)$. Consider the dual variable $y^\star \in \partial G(x^\star)$ such that $0 = \nabla F(x^\star) + y^\star$. Since, $y^\star \in \partial G(x^\star)$, $0 \in -x^\star + \partial G^*(y^\star)$ using $\partial G^* = (\partial G)^{-1}$. Finally,
$$\begin{bmatrix} 0 \\ 0 \end{bmatrix} \in \begin{bmatrix} \nabla F(x^\star) & +y^\star \\ -x^\star & +\partial G^*(y^\star) \end{bmatrix}. \tag{64}$$
Consider the set valued maps
$$B : (x,y) \mapsto \begin{bmatrix} \nabla F(x) \\ 0 \end{bmatrix},$$
and
$$A : (x,y) \mapsto \begin{bmatrix} y \\ -x & +\partial G^*(y) \end{bmatrix},$$
where we used vector notation. The maps $A$ and $B$ are maximal monotone operators (note that $B$ is the gradient of $(x,y) \mapsto F(x)$ and $A$ was used in Section 4). Inclusion (64) can be rewritten as
$$0 \in (A+B)(x^\star, y^\star). \tag{65}$$
In order to solve (65), one can apply the Forward Backward algorithm
$$P(x^{k+1/2} - x^k) = -\gamma B(x^k), \quad P(x^{k+1} - x^{k+1/2}) \in -\gamma A(x^{k+1}), \tag{66}$$
for a well chosen $P \in \mathbb{R}_{++}^{d \times d}$. As above, we take $P : (x,y) \mapsto x$. Although the matrix $P$ is only semi-definite positive, we showed in Lemma (4) that $x^{k+1} = \text{prox}_{\gamma G}(x^{k+1/2})$. Hence, the primal dual Forward Backward algorithm (66) is equivalent to the proximal gradient algorithm (63).

Moreover, the proof technique used for Theorem 5 can be adapted to analyze the proximal gradient algorithm as a primal dual algorithm. The complexity result obtained for the proximal gradient algorithm with this approach is suboptimal. However, the derivation of this complexity result sheds some light on PSGLA.

First, using the (strong) convexity of $F$,
$$\begin{aligned} \|x^{k+1/2} - x^\star\|^2 &= \|x^k - x^\star\|^2 + \gamma^2 \|\nabla F(x^k)\|^2 - 2\gamma \langle \nabla F(x^k), x^k - x^\star \rangle \\ &\leq (1 - \gamma\lambda_F)\|x^k - x^\star\|^2 + \gamma^2 \|\nabla F(x^k)\|^2 - 2\gamma \left(F(x^k) - F(x^\star)\right) \\ &\leq (1 - \gamma\lambda_F)\|x^k - x^\star\|^2 + \gamma^2 \|\nabla F(x^k)\|^2 - 2\gamma \left(F(x^{k+1/2}) - F(x^\star)\right) \\ &\quad - 2\gamma \left(F(x^k) - F(x^{k+1/2})\right). \end{aligned}$$
Using the smoothness of $F$,
$$F(x^{k+1/2}) - F(x^k) \leq \langle \nabla F(x^k), x^{k+1/2} - x^k \rangle + \frac{L}{2}\|x^{k+1/2} - x^k\|^2 = -\gamma\left(1 - \frac{\gamma L}{2}\right)\|\nabla F(x^k)\|^2.$$
Therefore,
$$\|x^{k+1/2} - x^\star\|^2 \leq \|x^k - x^\star\|^2 - \gamma^2\left(1 - \gamma L\right)\|\nabla F(x^k)\|^2 - 2\gamma\left(F(x^{k+1/2}) - F(x^\star)\right). \tag{67}$$
Inequality (67) is analogue to Lemma 7. Moreover, using Lemma 4,
$$\begin{aligned} \|x^{k+1} - x^\star\|^2 &\leq \|x^{k+1/2} - x^\star\|^2 \\ &\quad - 2\gamma\left(G^*(y^{k+1}) - G^*(y^\star) - \langle y^{k+1}, x^\star \rangle + \langle y^\star, x^{k+1/2} \rangle\right) \\ &\quad - \gamma(\lambda_{G^*} + \gamma)\|y^{k+1} - y^\star\|^2 + \gamma^2 \|y^\star\|^2, \end{aligned}$$
where $y^{k+1} = \text{prox}_{G^*/\gamma}(x^k/\gamma)$. Summing the two last inequality, and using $\gamma \leq 1/L$,
$$\begin{aligned} \|x^{k+1} - x^\star\|^2 &\leq (1 - \gamma\lambda_F)\|x^k - x^\star\|^2 - \gamma(\lambda_{G^*} + \gamma)\|y^{k+1} - y^\star\|^2 \\ &\quad - 2\gamma\left(\mathscr{L}(x^{k+1/2}, y^\star) - \mathscr{L}(x^\star, y^{k+1})\right) + \gamma^2 \|y^\star\|^2, \end{aligned} \tag{68}$$
where $\mathscr{L}(x,y) = F(x) - G^*(y) + \langle x, y \rangle$ is the Lagrangian function and $\mathscr{L}(x^{k+1/2}, y^\star) - \mathscr{L}(x^\star, y^{k+1})$ is the duality gap. The last inequality is similar to Theorem 5.

**Remark 1.** With slight modifications of the derivations above, one can get the better result
$$\|x^{k+1} - x^\star\|^2 \leq (1 - \gamma\lambda_F)\|x^k - x^\star\|^2 - 2\gamma \left( \mathscr{L}(x^{k+1}, y^\star) - \mathscr{L}(x^\star, y^{k+1}) \right).$$
However, the proof technique would not adapt to Langevin algorithm.

**Remark 2.** Similarly to the result of Theorem 5, $x^\star, y^\star$ can be replaced by any $\bar{x}, \bar{y}$. The proof technique does not use specific properties of $x^\star, y^\star$, as being primal dual optimal. Primal dual optimality of $x^\star, y^\star$ is only needed to prove that the duality gap is nonnegative.

## D  Generalization to a stochastic three operators splitting

In order to cover more applications, for instance involving *several Lipschitz proximable terms* in the potential, we quickly generalize the results of Section 5. Our primal dual framework can be plugged to the results of [27] instead of [19], leading to an extension of Section 5.

Consider the task of sampling from $\mu^\star \propto \exp(-V)$, where
$$V(x) = \mathbb{E}(f(x, \xi)) + \mathbb{E}(r(x, \xi)) + G(x). \tag{69}$$
We assume the following.

**Assumption 6.** For every $x \in \mathsf{X}$, $r(x, \xi)$ is integrable and $R(x) := \mathbb{E}_\xi(r(x, \xi))$. Moreover, $r(\cdot, \xi) \in \Gamma_0(\mathsf{X})$ a.s. Finally, there exists $M \geq 0$ such that for every $x \in \mathsf{X}$, $\mathbb{E}_\xi(\|\partial^0 r(x, \xi)\|^2) \leq M^2$.

Assumption 6 holds *e.g.* if $r(x, \xi)$ is $\ell(\xi)$-Lipschitz continuous and $\mathbb{E}_\xi(\ell^2(\xi)) < \infty$, since $\mathbb{E}_\xi(\|\nabla^0 r(x, \xi)\|^2) \leq \mathbb{E}_\xi(\ell^2(\xi))$. By replacing $F$ by $F + R$ in Section 3, Theorem 3 still hold with the Lagrangian function
$$\mathscr{L}(\mu, y) := \mathcal{E}_F(\mu) + \mathcal{E}_R(\mu) + \mathcal{H}(\mu) - \mathcal{E}_{G^*}(\nu) + \mathbb{E}\langle x, y\rangle, \tag{70}$$
where $x = T^\mu_{\mu^\star}(x^\star)$. Note that $\mathrm{dom}(R) = \mathsf{X}$, hence $R$ is differentiable a.e. using [24, Theorem 25.5]. In order to sample from $\mu^\star$, the stochastic proximal Langevin algorithm (SPLA) [27] is written
$$x^{k+1} = \mathrm{prox}_{\gamma G} \left( \mathrm{prox}_{\gamma r(\cdot, \xi)} \left( x^k - \gamma \nabla_x f(x^k, \xi^{k+1}) + \sqrt{2\gamma} W^{k+1} \right) \right). \tag{71}$$

SPLA recovers PSGLA by taking $R \equiv 0$. Moreover, SPLA is analyzed in [27] only in the case where $G$ also satisfies Assumption 6 (and hence $\mathrm{dom}(G) = \mathsf{X}$) which is stronger than assuming 4. In this section, we denote
$$x^{k+1/2} := \mathrm{prox}_{\gamma r(\cdot, \xi)} \left( x^k - \gamma \nabla_x f(x^k, \xi^{k+1}) + \sqrt{2\gamma} W^{k+1} \right), \tag{72}$$

and $\mu^{k+1/2}$ its distribution. We shall prove the following extension of Theorem 5 assuming that $G$ satisfies the Assumption 4. This extension of Theorem 5 leads to an extension of Corollary 6 providing complexity results for SPLA similar to PSGLA.

**Theorem 8.** Let Assumptions 1, 2, 3, 5 and 6 hold true. If $F$ is $\lambda_F$-strongly convex and $G$ is $1/\lambda_{G^*}$-smooth, then for every $\gamma \leq 1/L$,
$$W^2(\mu^{k+1}, \mu^\star) \leq (1 - \gamma\lambda_F)W^2(\mu^k, \mu^\star) - \gamma(\lambda_{G^*} + \gamma)W^2(\nu^{k+1}, \nu^\star)$$
$$- 2\gamma \left( \mathscr{L}(\mu^{k+1/2}, y^\star) - \mathscr{L}(\mu^\star, y_\star^{k+1}) \right) + \gamma^2 C, \tag{73}$$

where $C := \int_{\mathrm{int}(D)} \|\nabla G(x)\|^2 d\mu^\star(x) + 2(Ld + \sigma_F^2 + M^2)$ and $y_\star^{k+1} := \mathrm{prox}_{G^*/\gamma}(x_\star^{k+1/2}/\gamma) \sim \nu^{k+1}$, where $x_\star^{k+1/2} := T^{\mu^{k+1/2}}_{\mu^\star}(x^\star)$.

Before proving Theorem 8, we recall the following consequence of [27, Theorem 1], which generalizes Lemma 7.

**Lemma 9** ( [27]). Let Assumptions 1, 2, 5 and 6 hold true. Then, if $F$ is $\lambda_F$-strongly convex, for every $\gamma \leq 1/L$,
$$W^2(\mu^{k+1/2}, \mu^\star) \leq (1 - \gamma\lambda_F)W^2(\mu^k, \mu^\star) + 2\gamma^2(Ld + \sigma_F^2 + M^2)$$
$$- 2\gamma \left( \mathcal{E}_F(\mu^{k+1/2}) + \mathcal{E}_R(\mu^{k+1/2}) + \mathcal{H}(\mu^{k+1/2}) - \mathcal{E}_F(\mu^\star) - \mathcal{E}_R(\mu^\star) - \mathcal{H}(\mu^\star) \right). \tag{74}$$

*Proof.* Apply [27, Theorem 1] by taking $G_2 \equiv \ldots \equiv G_n \equiv 0$ and noting that the KL term in [27, Equation 3] is equal to $\left(\mathcal{E}_F(\mu^{k+1/2}) + \mathcal{E}_R(\mu^{k+1/2}) + \mathcal{H}(\mu^{k+1/2}) - \mathcal{E}_F(\mu^\star) - \mathcal{E}_R(\mu^\star) - \mathcal{H}(\mu^\star)\right)$ using our notations, see Equation (15). □

We now prove Theorem 8, similarly to Theorem 5.

The main tool for the proof is Lemma 4. Replace $x$ by $x_\star^{k+1/2} \sim \mu^{k+1/2}$ in (26). Then $y' = y_\star^{k+1} \sim \nu^{k+1}$ and $\mathrm{prox}_{\gamma G}(x_\star^{k+1/2}) \sim \mu^{k+1}$. Therefore,

$$W^2(\mu^{k+1}, \mu^\star) \leq \mathbb{E}(\| \mathrm{prox}_{\gamma G}(x_\star^{k+1/2}) - x^\star \|^2), \quad W^2(\nu^{k+1}, \nu^\star) \leq \mathbb{E}(\| y_\star^{k+1} - y^\star \|^2).$$

Consequently, taking expectation in (26) we get

$$W^2(\mu^{k+1}, \mu^\star) \leq W^2(\mu^{k+1/2}, \mu^\star) - \gamma(\lambda_{G^*} + \gamma) W^2(\nu^{k+1}, \nu^\star) + \gamma^2 \int \|y\|^2 d\nu^\star(y)$$
$$- 2\gamma \left( \mathcal{E}_{G^*}(\nu^{k+1}) - \mathcal{E}_G^*(\nu^\star) - \mathbb{E}\langle y_\star^{k+1}, x^\star \rangle + \mathbb{E}\langle y^\star, x_\star^{k+1/2} \rangle \right).$$

Combining with Lemma 9, we get the result.