[Reviews · NeurIPS 2020]

Review 1

Summary and Contributions: This paper considers and analyses proximal stochastic gradient Langevin algorithm which consists of regular ULA step on the differentiable function followed by a proximal step. This algorithm is relevant when G is not differentiable. The paper establishes a convergence result in Wasserstein-2 which was not established before.

Strengths: The paper consists of analysis which looks sound and the interpretation in the space of measures can lead to new results for related methods.

Weaknesses: The analysis is fairly technical and there is less emphasis on empirical performance in the main paper. This would limit audience of the paper.

Correctness: Yes. The analysis is done in a valid and novel way.

Clarity: There is a lot of auxiliary notation introduced and the paper is fairly technical. It may not be accessible to the most of the NeurIPS community.

Relation to Prior Work: Yes, previous contributions are properly cited and discussed.

Reproducibility: Yes

Additional Feedback: Post rebuttal: The authors addressed my comments. Therefore, I keep my score as 'accept' but not higher as I think the clarity of the writing should be improved. -- This paper considers an important scheme for sampling from probability measures of the form exp(-F-G) using the structure of F and G. When G is nonsmooth and proximable, using proximal maps lead to much faster convergence compared to using subgradients in the optimization case. It is therefore an important problem to investigate the sampling analogue of this scheme which is the topic of this paper. As mentioned, some previous work has been done on this problem, but this paper presents an approach that is most general (in terms of G being supported on a more general set) to date. (1) I am confused about Assumption 3: The main motivation of the paper seems to be tackling the case where G is non-smooth but herein the paper talks about its derivatives. What is the definition of \nabla G? Does theory require its differentiabillity? In this case, the results would be less applicable than claimed. (2) I am more familiar with the standard proximal gradient techniques for optimization. In principle, eq. (3) can be seen as an optimizer: One can merge the noise vector W into the gradient noise if the gradient is subsampled and stochastic. In this case, the results of this paper imply that these optimizers can be seen as samplers (which is suggested also before). Can the authors elaborate with a couple of sentences the relationship between eq. (3) and more traditional proximal-gradient optimizers? This would also relate to applications of this result to optimization in the non-convex case. (3) Proximity operators cannot be implemented in general and are usually approximated. Would it be easy to extend these results to the inexact proximal mappings with bounded errors? (4) The main results are interesting and it is clearly shown that (e.g. in eq. (30)) the convergence of the PSGLA in W-2 distance. This result is also similar to the stochastic proximal gradient result in Euclidean norm for the strongly convex case and interesting. How important is the convexity of F in Corollary 6? In other words, would it require substantially different techniques to extend this result to nonconvex cases? If so, why? 


Review 2

Summary and Contributions: This paper proposes a primal-dual interpretation of Proximal Stochastic Gradient Langevin Algorithm. A parallel with this type of algorithm in optimisation is drawn, shedding light on the relations between optimisation and sampling algorithms. By making use of the interpretation of sampling as optimisation in the space of measures, the authors define a new duality gap specific to sampling, prove an equivalent to strong duality in terms of this quantity, and use it to show overall convergence of the algorithm.

Strengths: The theoretical framework provides progress in understanding the parallel between optimisation and sampling algorithms. Numerical results show the effectiveness of the studied algorithm.

Weaknesses: The studied algorithm is known, and this new analysis does not really yield improved convergence guarantees (at the exception of the square Lipschitz constant being replaced by the average of the square norm of the gradient compared to [18]). No detail about the dimension dependence in the convergence guarantees is given. This would be good to add, since such methods are usually aimed to be applied to large dimensional problems.

Correctness: The claims and methodology are correct.

Clarity: The paper is well written, although some notations are a bit confusing. In particular, you define x^\star as being the map (x,y) -> x, making it a function, but also say x^\star \sim \mu^\star, making it a vector in R^d...

Relation to Prior Work: Relations and differences to prior work is made clear.

Reproducibility: Yes

Additional Feedback: "To the best of our knowledge, inequality (29) that holds when F is just 303 convex is new, even in the particular case where G is Lipschitz": Actually, recent work [1] already showed convergence of ULA in 2-Wasserstein distance in the case where the target distribution is weakly log-concave. What would your framework boil down to in the specific case of smooth constrained sampling, i.e., when G is the indicator function of some convex set? In this case, PSGLD is simply the Projected Langevin Algorithm. What kind of guarantee do you obtain? We would have that lambda_{G^*}=0, meaning that equation (29) does not yield convergence in W2. However, equation (28) would still yield convergence in terms of this new duality gap. However, it's hard to make sense of the convergence guarantee given in terms of this duality gap, even in this specific setting. Minor: 142 "nose" [1] Rolland P., Eftekhari A., Kavis, A., Cevher, V, "Double-Loop Unadjusted Langevin Algorithm", ICML, 2020 ======= Post-rebuttal edit: The responses from the authors were satisfactory. The main contribution of this work is indeed to enlarge the class of distributions from which PSGLA is able to sample. I am thus willing to raise my score to 7.


Review 3

Summary and Contributions: This paper studies the performance of sampling w.r.t log concave probability distributions, these distributions involve in the exponential term is a sum of a two terms: a convexe smooth one and a non smooth convex and possibly infinite one. The idea developed here is to generalize the Langevin sampler (only possible for smooth pdfs as it involves gradient) to this particular case while taking advantage of the proximal approach allowed by the penalty term. The authors reformulate the task of sampling to define a duality gap for the minimization of the KL divergence, to represent PSGLA as a primal dual  stochastic Forward Backward algorithm. Using this framework, the authors prove new complexity results for PSGLA in the case mentioned above.

Strengths: The paper technically sounds. Although technical, the results are clearly stated and the elements for understanding the whole process of mind are given along the paper progression. The explanations of each assumption make sense and are very useful and interesting. The results improve the state of the art theoretical resutls.

Weaknesses: The only weakness is that the authors mention numerical experiments for sampling from a distribution supported by a half space of matrices but this part does not appear in the paper. Some notations may either not be defined (\mathbb{V}_\xi for example, although I guess it stands for the variance of f) or misleading (\nabla f(.,s) \equiv \nabla F for example where one is replaced by the other and not equal to).

Correctness: The paper looks correct.

Clarity: Although technical, the paper is well written. Anyway, this may be a paper for experts only.

Relation to Prior Work: Yes, the bibliography is well figured out and relevant.

Reproducibility: Yes

Additional Feedback:


Review 4

Summary and Contributions: The authors consider the problem of sampling from a log-concave distribution. Specifically, they study a variant of unadjusted Langevin algorithm, that is suitable for distributions whose support may not be the whole space. Specifically, they provide a derivation of an algorithm (proximal stochastic gradient Langevin algorithm), and present results on its convergence rate.

Strengths: The work is potentially of interest for computational Bayesian statistics, since it extends the class of distributions that one can sample from, if one wanted to use some form of a Langevin based sampling algorithm without a correction stage. It would also be of interest to researchers working on sampling schemes based on Langevin diffusion in general.

Weaknesses: Practical significance of the discussed method is not clear to me. Normally, this could be addressed via numerical experiments, but they seem to be rather simple, for a paper of this complexity. Also, I don't think non-experts in the area will get much out of the paper because it does not provide much intuition. Despite a 10-page appendix, the paper does not really help clarify the connection between convex optimization and the problem of sampling from a log-concave distribution. To be fair, there is a lot of preliminary knowledge that needs to be covered, and this takes up most of the paper -- what I would have liked to see is pointers for researchers familiar with convex optimization to help understand the underlying convex mechanism.

Correctness: I have minor questions, but apart from that, I have not come across gross mistakes regarding convex optimization.

Clarity: The authors could have provided more intuition in the main text, so as to at least draw people with different backgrounds to the problem and techniques in the paper. However, in its current form, for non-experts, I think the paper would serve mostly as a pointer to references.

Relation to Prior Work: Yes, the authors have described in detail various recent work on the subject to clarify the contribution.

Reproducibility: Yes

Additional Feedback: - line 21 : it is noted that (1) produces samples (asymptotically) from the target distribution. Are we assuming that \gamma decays to zero? If not, I think the Langevin algorithm does not produce samples from exp(-V) -- In fact, refs [16],[19] provide bounds for this error. - For eqn [3], is the reference [18] the correct one? - In the one-dimensional example, shouldn't we compare against MYULA with a small gamma? When I take gamma = 1e-3, or 1e-4 in the code you provided, the distribution from MYULA is closer to the target -- how do you interpret this? - line 170 : Should it be T : X \to Z?, since we are applying \mu to T^{-1}(A)? - line 198 : Do we also have some restrictions on \mu' for (12) to hold? In general, how similar is geodesic convexity to regular convexity -- should we think of them as practically the same, or are there important differences that prevent application of results from convex analysis in Hilbert spaces? - I'm confused by statement of Thm 2 : I understand w is a random variable (with range R), since it's in the subdifferential. Then, isn't the lhs of (16) a scalar, and the rhs in R^d? - line 237 : how can we have V=0 and \nabla V \neq 0? Please expand/clarify the discussion leading to eqn (17). - line 244 : What is the meaning of G^* in this context? Does the conjugate have any probabilistic interpretation in this context? ***** post rebuttal update I read the other reviews and the author rebuttal. I thank the authors for their responses. I still think the paper would of interest to experts in this area, but could be made more accessible to a wider interested audience. My general score has not changed.


Review 5

Summary and Contributions: This paper provides a primal-dual view of the PSGLA algorithm under a general setting where the support of \mu^* might not lie in X. The authors establish the strong duality of the corresponding Lagrangian. Based on the primal-dual optimality, the authors view PSGLA as a stochastic Forward-Backward algorithm and establish a refined convergence guarantee of PSGLA under the general setting.

Strengths: 1. The authors propose a primal-dual formulation of proximal Langevin algorithm and justify the strong duality, which is novel to me. 2. Based on the primal-dual formulation, the authors reformulate PSGLA into a stochastic Forward-Backward algorithm and provide analysis to PSGLA accordingly. 3. The authors refine the previous convergence results of PSGLA under a weaker condition.

Weaknesses: 1. The discussion of Forward-Backward representation is relatively sloppy. The definition of Lagrangian also lacks intuition. 2. The fact that the support \mu^* might not align (which extends the previous results and is a major contribution) is scarcely touched. It would be helpful for the author to discuss how the primal-dual formulation tackles the challenge incurred by the infinite value of G.

Correctness: The claims are proved and are intuitively correct to me.

Clarity: The paper is in general well written, though clarity in the discussion of results could be improved.

Relation to Prior Work: A comprehensive discussion and comparison of this work with previous works are included.

Reproducibility: Yes

Additional Feedback: Many of my concerns are on the organization of the paper. I would suggest more discussion in the Forward-Backward method (for instance, some discussion similar to that in Appendix C.2) and comparison of the convergence result in Corollary 6 with previous works. I would also suggest a more succinct Background section. Some results presented in the background does not contribute to the discussion of main results, and I would suggest them moved to the Appendix section.

[Author Response · NeurIPS 2020]

We thank Reviewers (R) 1, 2, 3, 4 and 5 (who gave us marks 7, 6, 8, 6, and 6, respectively) for their pertinent remarks.

**R2+R4 (also R1+R5): Main contribution.** We recall our main contribution. Recall that $\mathrm{prox}_G$ generalizes the
projection onto convex set in a way that $\mathrm{prox}_G(x) \in \mathrm{dom}(G)$. Importantly, in PSGLA *the iterates are feasible*:
$x^{k+1} \in \mathrm{dom}(G) = \mathrm{supp}(\mu^\star)$, contrary to alternative methods. The PSGLA method proposed in this paper *extends in*
*a natural way* particular cases that were considered in the literature, namely i) the case where $G$ is Lipschitz [18, Sec
4.2] (in particular $\mathrm{supp}(\mu^\star) = \mathsf{X}$) and ii) the case where $G$ is the indicator of a convex compact set [9] (in particular
$\mathrm{supp}(\mu^\star) \neq \mathsf{X}$), in which case PSGLA has a high complexity $\mathcal{O}(1/\varepsilon^{12})$. All the other cases, where $G$ is a general
convex l.s.c, potentially with a domain $\mathrm{dom}(G) \neq \mathsf{X}$ (for instance the $G$ considered in the experiments l.456) where
not analyzed before. *Our main contribution is to analyze PSGLA in these new cases*. This is a challenging problem:
we had to develop new mathematical tools (e.g. the duality gap arising from the primal dual interpretation of PSGLA)
for the analysis. Using these tools, we obtained surprising results: although $G$ can have a domain ($\mathrm{supp}(\mu^\star) \neq \mathsf{X}$),
the complexity of PSGLA in these new cases is basically the same as in the case where $G$ is Lipschitz ($\mathcal{O}(1/\varepsilon^2)$).

**R1+R3+R4: Numerical experiments.** Although our paper is mainly of a theoretical nature, we provided detailed
numerical experiments + the associated code. We will use the 9th page to move some experiments to the main text if
our paper is accepted. We do not believe that our experiments are too simple as R4 says. We considered a sampling
problem in a multidimensional half space of matrices relevant in the field of random matrices. The function $G$ is given
in l.456 and the computation of $\mathrm{prox}_G$ in closed form relies on recent results on proximity operators. Particular cases
of PSGLA considered previously are not able to tackle this sampling problem. The main message of the experiment
section is: PSGLA only produces iterates in the support of $\mu^\star$, contrary to alternative methods.

**R1:** 1) $G$ is a general convex l.s.c. but using [24, Th 25.5], $G$ is almost surely differentiable on its domain, therefore
the integral is well defined. 2) SGD can be written $x^{k+1} = x^k - \gamma \nabla F(x^k) + \gamma w^{k+1}$, where $w^{k+1}$ is a martingale
increment. In Langevin the noise $W^k$ is scaled by $\sqrt{\gamma}$ instead. There is more noise in Langevin, that's why Langevin
explores the whole support of $\mu^\star$, whereas SGD converges to $\arg\max \mu^\star$. 3) Yes, and we did it! In Appendix D, we
used (cheap) stochastic proximity operators instead of full proximity operators and showed that the convergence rates
remain unchanged. Another approach (that we will acknowledge) could be to adapt to our setting the proof technique of
Pesquet and Combettes in a series of papers on primal dual optimization with approximate proximity operators. Note
however that many proximity operators can be computed in closed form (hinge loss, logistic loss, many penalizations,
etc., see also the experiments l.456) thanks to research efforts on this topic, see proximity-operator.net. 4) Convexity is
needed to prove that the duality gap is nonnegative (Th3) which is fundamental in our approach (We iterated (27)). For
nonconvex cases, see [30,33] that made nice connections between nonconvex Langevin and nonconvex optimization.

**R2:** 1) We think that R2 has missed a key part of our main contribution, see above. 2) In the literature on Langevin
algorithm, complexity results are indeed often expressed in terms of the parameters of the problem like the Lipschitz
constants and $d$. If $G$ is $M$-Lipschitz, [18, discussion after Corollary 18] gives a complexity result for PSGLA in terms
of $\sigma_F^2, L, M, d$ by using an approach similar to [[1, Lemma 5]]. Although the constants $\sigma_F^2, L, M$ implicitly depend
on $d$, they have an explicit meaning. Using [[1, Lemma 5]] in our case, we can obtain the same complexity result as in
[18], but by replacing $M^2$ by $I := \int \|\nabla G\|^2 d\mu^\star$. We will acknowledge that $I$ is a bit less intuitive than $M$, but $\sqrt{I}$ can
be seen as a generalized Lipschitz constant since $I \leq M^2$ if $G$ is $M$-Lipschitz. 3) $x^\star$ is a measurable map $\Omega \to \mathsf{X}$, i.e.,
a random vector (see l.246). Since $x^\star : (x, y) \mapsto x$ and $\Omega = \mathsf{X}^2$ is endowed with $\pi^\star$, the distribution of $x^\star$ is the first
marginal of $\pi^\star$, i.e., $\mu^\star$ (such construction is sometimes used in probability theory). 4) l.303: we will acknowledge
[[1]] (we did not know). 5) The meaning of the convergence of the duality gap to zero is an important interesting
question that is the subject to further research. We are specifically interested in understanding its relationship with
weak or KL convergence. It is also a difficult question, and we don't have a clear answer for projected Langevin. In
this paper, we focused on the convergence in $W_2$ in the case where $F$ is strongly convex and $G$ general convex l.s.c,
in order to cover new cases.

**R1+R3:** Thank you for your positive feedback. We will use the 9th page to move some experiments to the main paper.

**R1+R4+R5 (also R3): Intuition.** Regarding the FB, everything is made rigorous in Lemma 4. We provided a whole
Appendix (C) to provide intuition on the relationship on primal dual convex optimization, FB and sampling, and how
we had the idea of using a primal dual view to obtain our complexity results. Notably, our Lagrangian is similar to the
one used in optimization, see C.2. We will go further by moving some material from C.2 to the introduction.

**R4:** l.21, l.170: we corrected. l.198: (12) is valid only for any $\mu'$, but only using $T_\mu^{\mu'}$ the *optimal* pushforward. This
is the main difference with convexity, that makes things more challenging. Th2: $w$ is a (random) vector because the
elements of the subdifferential are (random) vectors (as in optimization), see l.195. Th2: If $V \equiv 0$, $w\rho = \nabla\rho$ i.e.
$w = \nabla \log(\rho)$. Then we take $\rho = \exp(-V)$ which gives the result. $\gamma$: There is a trade-off: if $\gamma$ is small, MYULA
(or any Langevin algorithm) is more precise but slower to reach its invariant distribution. That's why we compared
MYULA vs PSGLA at equal learning rate $\gamma$.

[Meta-Review · NeurIPS 2020]

This paper presents a new analysis for proximal stochastic gradient Langevin algorithm. All the reviewers recognized the intellectual merits with minor concerns. I would suggest the authors to carefully revise their paper based the reviewers' comments.